# REPRESENTATION SHATTERING IN TRANSFORMERS: A SYNTHETIC STUDY WITH KNOWLEDGE EDITING

## ABSTRACT

Knowledge Editing (KE) algorithms alter models' weights to perform targeted updates to incorrect, outdated, or otherwise unwanted factual associations. To better identify the possibilities and limitations of these approaches, recent work has shown that applying KE can adversely affect models' factual recall accuracy and diminish their general reasoning abilities. While these studies give broad insights into the potential harms of KE algorithms, e.g., via performance evaluations on benchmarks, we argue little is understood as to *why* such destructive failures occur. Is it possible KE methods distort representations of concepts beyond the targeted fact, hence hampering abilities at broad? If so, what is the extent of this distortion? Motivated by such questions, we define a novel synthetic task wherein a Transformer is trained from scratch to internalize a "structured" knowledge graph. The structure enforces relationships between entities of the graph, such that editing a factual association has "trickling effects" on other entities in the graph (e.g., altering X's parent is Y to Z affects who X's siblings' parent is). Through evaluations of edited models and analysis of extracted representations, we show that KE inadvertently affects representations of entities beyond the targeted one, distorting relevant structures that allow a model to infer unseen knowledge about an entity. We call this phenomenon *representation shattering* and demonstrate that it results in degradation of factual recall and reasoning performance more broadly. To corroborate our findings in a more naturalistic setup, we perform preliminary experiments with pretrained GPT-2-XL and Mamba models, reproducing the representation shattering effect therein as well. Overall, our work yields a precise mechanistic hypothesis to explain why KE has adverse effects on model abilities.

## 1 INTRODUCTION

Large language models (LLMs) have led to unprecedented advances in several domains (Gemini Team, 2023; Bubeck et al., 2023; Touvron et al., 2023; Thoppilan et al., 2022; Chowdhery et al., 2022; Qin et al., 2023; Chen et al., 2021; Ahn et al., 2022; Driess et al., 2023). However, the static nature of their training pipelines implies that as our world evolves, models' internalized knowledge can become incorrect or outdated. To address this, recent work has proposed several protocols for knowledge editing (KE), wherein the goal is to minimally and precisely alter model weights such that only the targeted information (and its relevant associations) are updated, but all unrelated information remains (ideally) unaffected (Mitchell et al., 2022; Meng et al., 2022a; 2023; Dai et al., 2021; Cheng et al., 2023; De Cao et al., 2021; Sinitsin et al., 2020).

Despite significant work on the topic, it still remains unclear precisely what effects KE should have on a model. For example, assume you edit the fact that "*Michael Jordan* won the 1998 NBA MVP" to "*Reggie Miller* won the 1998 NBA MVP", then what should the impact of such an edit be? Should the model now believe Michael Jordan and the Chicago Bulls never reached the NBA finals in 1998? Should it perhaps believe Reggie Miller was on the Chicago Bulls? Should the pop quote "Be like Mike" (Wikipedia, 2024) now become "Be like Reggie"? As Hofweber et al. (2024); Hase et al. (2024) argue, it is difficult to design clear, well-defined answers for such questions. Motivated by this, recent work has started investigating precisely what effects KE actually has on the model (Hoelscher-Obermaier et al., 2023; Li et al., 2023b; Lynch et al., 2024). For example, Cohen et al. (2023) demonstrate that knowledge beyond the edited fact can often be impacted in a detrimental manner, such that the model begins to have an incoherent understanding of the world; Gupta et al. (2024) demonstrate unrelated facts are often forgotten by the model post-editing; and

Figure 1: ***Representation shattering* explains why knowledge editing can sometimes degrade models' general capabilities.** *(a)* Prior works finds that editing facts, e.g., the president of the US, can sometimes harm general abilities of LLMs (figure reproduced from Gu et al. (2024)). *(b)* We introduce a synthetic data generation process defined by a knowledge graph containing ring-shaped geometries. When we train a model on our synthetic data, we observe that the model's internal representations mirror the ring structure of the underlying data generation process. We then explain the post-edit degradation of the model's capabilities by uncovering the "shattering" of latent representations. In the example provided, while the edit successfully changes the fact (Entity #124 is to the right of Entity #123), the model's performance metrics drop after the knowledge edit.

Gu et al. (2024) show that KE can harm broader reasoning capabilities beyond mere factual recall. While these works clearly demonstrate the detrimental impacts of editing on a model, they still leave open the question precisely *why* such harms occur—at a mechanistic level, how are model representations impacted such that a broad set of knowledge and capabilities in a model are heavily distorted once an edit occurs?

**This work.** To address the questions above, we aim to develop a mechanistic understanding of the impact of KE on a model's internals. For this purpose, we argue we must solve two problems: (i) identify how a model expresses knowledge about some predefined set of entities in its representations, and (ii) investigate how this mechanism is affected as we apply KE to alter a fact corresponding to a subset of the entities. Instead of attacking a complicated system that may be difficult to interpret (e.g., an off-the-shelf LLM), we take inspiration from a multitude of recent papers that establish synthetic abstractions of the target system and develop precise hypotheses as to why the phenomenon-in-question occurs (Allen-Zhu & Li, 2023c;a;b; Okawa et al., 2023; Chan et al., 2022; Li et al., 2023a; Lubana et al., 2024). Specifically, we define a data-generating process that yields entities arranged in a *structured* knowledge graph. This structure is defined via use of a predefined set of relations that *locally* constrain how entities relate to each other (similar to parent-child relations). Given enough entities and relations, such local constraints manifest a broader *global* structure in the knowledge graph. Performing traversal over the nodes of this knowledge graph, we get sequences that can be used as "strings" to train a Transformer on. As we show, this protocol leads to the model precisely encoding the structure of the graph in its latent representations. However, when KE is applied to edit either incorrectly learned facts or insert counterfactual knowledge (using the method proposed by Meng et al. (2022a)), we find latent representations are heavily distorted and the graph structure completely destroyed—we call this phenomenon **representation shattering**. Interestingly, this phenomenon manifests in proportion to how far the proposed edit moves a given node from its current location to a new location in the graph (defined via edge distance). We thus hypothesize representation shattering underlies the detrimental effects of KE on a pretrained model's factual and reasoning capabilities at broad. Overall, we make the following contributions in this work.

- **Structured Knowledge Graphs as a Toy Setting for Investigating Impact of KE.** We propose use of a structured knowledge graph wherein entities (nodes) are connected to each other via predefined local constraints (relations) that manifest into a broader, global structure in the graph (see Sec. 3). Training Transformers on strings (path traversals) from the graph, we find model representations precisely encode the global structure of the graph. This allows us to assess the impact of KE at a more mechanistic level, since distorting a fact now has global effects that can be precisely (and, in fact, visually) delineated by analyzing the model representations.

- **Representation Shattering as a Mechanistic Hypothesis to Explain Detrimental Effects of KE.** We find KE distorts latent representations for entities in the graph such that the global geometry learned during pretraining is, at times, completely destroyed—we call this phenomenon **representation shattering** and hypothesize it underlies the detrimental effects of KE on model capabilities observed in prior work (see Sec. 4). As we show, the extent of harm on latent representations turns out to be correlated to the amount an edit alters the graph from its original organization into the new, desired one.

- **Investigations with Off-the-Shelf LLMs.** Using pre-trained GPT2-XL and Mamba models, we provide evidence for our claims about representation shattering in more naturalistic settings. For one, we find real-world analogues to our synthetic knowledge graph structures (i.e., days of the week) and reproduce similar shattering phenomena in GPT2-XL and Mamba to what we observe in our toy setup (see Sec. 4.5). Additionally, we further reinforce the generality of our findings with preliminary replications of representation shattering under more complex knowledge graph geometries, such as trees (i.e. countries and their cities).

## 2 RELATED WORK

**Knowledge Editing.** Several protocols for knowledge editing (KE) have been proposed in recent work. Early work defined meta-learning based approaches (Sinitsin et al., 2020; De Cao et al., 2021; Mitchell et al., 2022) and established the broader desiderata for what properties a KE protocol should satisfy; e.g., ensuring facts unrelated to the target one are not hampered via the editing protocol. Building on work aimed at understanding how Transformers encode knowledge in their internals (Geva et al., 2020), modern KE protocols focus on defining closed-form operations that involve (i) localizing knowledge to specific components in a model (e.g., MLP layers) and (ii) defining operations to alter a factual association by assuming the fact is represented in a localized manner (Meng et al., 2022a; 2023).

**Evaluations of Knowledge Editing Methods.** As argued by Hase et al. (2024); Hofweber et al. (2024), the problem of KE is relatively ill-defined. Consequently, it is unclear that when we edit knowledge within a model, what effects said edits *should have* on other facts it may have internalized during training. Prior work has hence taken an alternative approach, primarily focusing on developing an empirical understanding of what the phenomenology of KE protocols is: e.g., if an edit is performed, how are counterfactual statements or unrelated facts affected. These works generally show that KE in fact has extreme detrimental effects on a model, e.g., hampering both its broader internalized knowledge and its reasoning abilities (Hase et al., 2023; Cohen et al., 2023; Hoelscher-Obermaier et al., 2023; Gupta et al., 2024; Gu et al., 2024). While the primary methodology used in such papers is to perform empirical benchmarking of a model that has undergone editing, we instead focus on a mechanistic analysis of how editing alters a model's representations (albeit primarily in a toy synthetic task) to yield the undesirable effects on model abilities.

**Explaining Models via Synthetic Tasks.** To disentangle the failures of KE methods from the failures of the models themselves, we argue for use of a more controllable and interpretable setup. Such a setup can help identify a concrete hypothesis for why KE has undesirable effects on the model, which we can then analyze in naturalistic settings by designing more precisely defined experiments. This methodology of designing toy, control tasks to investigate hypotheses for phenomenology of a neural network has yielded promising results in recent years, providing, e.g., a concrete hypothesis for how chain-of-thought reasoning aids model capabilities (Prystawski et al., 2024; Feng et al., 2023), models for emergent capabilities (Okawa et al., 2023; Lubana et al., 2024), existence of nonlinear representations (Engels et al., 2024), and failure modes for compositional generalization (Zhou et al., 2023).

## 3 FORMALZING KNOWLEDGE EDITING

Epistemology has grappled with the nature of knowledge for centuries (Chappell, 2005). In this work we adopt a humble, yet precise definition of knowledge based on structured knowledge graphs. A knowledge graph is used to represent how facts, entities, and relations are interlinked, giving rise to notions of consistency, coherency, and reasoning across different pieces of information. Using these definitions, we will define a synthetic data generation process on knowledge graphs, in order to systematically study knowledge editing in Transformers.

### 3.1 KNOWLEDGE GRAPHS

A knowledge graph consists of a collection of entities $X = \{x_i\}_{i=1}^n$, and a collection facts $F$ that relate different entities. For example, a graph defined on entities $X = \{$"Alice", "Bob", "Carol"$\}$ can encode the fact "Alice is the advisor of Bob" using the relation "advisor", represented as ("Alice", "advisor", "Bob").

**Definition 3.1 (Knowledge graph).** Formally, a knowledge graph $G = (X, R, F)$ consists of nodes $X$, relations $R$, and facts $F$, where each fact $f = (x_i, r, x_j) \in R$ is defined by a relation $r \in R$ between two entities $x_i, x_j \in X$.

A **relation sub-graph** corresponds to a sub-graph constructed by only considering facts that use relation $r$. For example $G_{\text{advisor}}$ is a sub-graph that specifies all facts for the relation "advisor". Every knowledge graph contains a collection of facts that can be inferred from the graph.

Related pieces of information such as "Alice's advisor was Bob" and "Bob's advisor was Carol" can be composed to form cohesive statements such as "Alice's advisor's advisor was Carol. To capture such statements, we define compositions of relations below. The composition of relations are essential to capture ripple effects that occur in the knowledge graph after an edit (Cohen et al., 2023) to a relation in $R$.

**Definition 3.2 (Composition of relations).** A composition of relations $\vec{r} = (r_1, r_2, \cdots, r_k) \in R^k$ with respect to knowledge graph $G$ is defined such that for every fact $f = (x_i, \vec{r}, x_j)$, there exists a collection of facts $\{(x_i, r_i, x_{i+1})\}_{i=1}^k$ for which $x_1 = x_i$ and $x_{k+1} = x_j$. In other words, any fact defined on the composition of relations has a corresponding set of facts defined on relations from $R$. Furthermore, the set of facts form a path in the knowledge graph such that the sequence of relations in the path are $r_1, r_2, \cdots r_k$.

## 3.2 CYCLIC GRAPHS: A DESCRIPTION OF THE ENTITIES AND RELATIONS

We study knowledge graphs where every relation sub-graph is a set of disjoint cyclic graphs, i.e., for every entity $x_i$ and relation $r$, there exists exactly one entity $x_j$ such that $(x_i, r, x_j) \in F$. We specifically choose a cyclic geometry as a global constraint on the graph structure since cycles are a common pattern that relate entities in natural language domains; e.g., see Fig. 2, where we show a 2D projection of representations from Llama-3.1-405B corresponding to months of a year and days of the week naturally organize in a cyclic fashion.

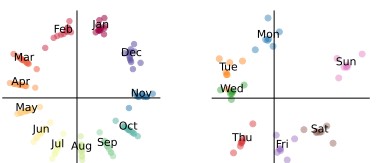

Figure 2: **Isomap projections of representations in Llama-3.1-405B (Fiotto-Kaufman et al., 2024).** The geometry of the data—for example, the cyclic nature of months or days—is often reflected in the representations learnt by language models. Similar representations can also be found in other models like GPT-2-Small and Mistral 7B (Engels et al., 2024).

Knowledge editing methods, e.g., ROME (Meng et al., 2022a; 2023), target a set of entities for which predefined facts are to be edited, while using another retain set of facts about said entities to help ensure relations beyond the targeted ones are not altered. A test set of facts are then used to evaluate how well the method worked. Motivated by this, we define a knowledge graph with 2048 entities (denoted by 1-2048) over which we define 3 cyclic orders (order I, II and III). The cyclic orders are generated using random permutations of the entities. We create 8 relations for each cyclic order totaling to 24 relations. The 8 relations correspond to the 1-hop, 2-hop, 3-hop and 4-hop neighbors in the clockwise and anti-clockwise directions in the cycle. The relations are named after a combination of the cyclic order (I, II, III), the neighbor's distance (between 1-4) and the neighbor's direction (Clockwise, Anti-clockwise). For instance, the relation "I_C2" denotes the 2-hop neighbor in the clockwise direction, with respect to cyclic order I." The 1-hop neighbor relation graphs (both clockwise and anti-clockwise) contain a single cycle, 2-hop relation graphs consist of 2 cycles, the 3-hop relation graph contains 1 cycle, while the 4-hop relation graph contains 4 cycles. The k-hop neighbor relations are related to each other by design, so any edit to one k-hop relation should be consistent with all other k-hop relations. An edit corresponds to changing a fact in the knowledge graph and can also be interpreted as changing an edge in the relation graph. For an illustrative example, see Fig. 3.

Depending on the fact being edited, the 3 cyclic orders are used to define the edit sub-graph, the retain sub-graph, and the test sub-graph. Why do we create 3 cyclic orders? The knowledge editing method targets **edit sub-graph relations**. The facts based on edit relations are then tested to check if a knowledge edit was successful. The **retain sub-graph relations** are used by the knowledge editing algorithm to minimize changes to unrelated relations, but no edits are made to facts that use these relations. The **test sub-graph relations** are used to define facts that are neither directly edited, nor used by the knowledge editing algorithm. The relations are used to evaluate whether unrelated

facts remain unchanged after a knowledge edit. We note that relations for all 3 sub-graphs are seen during pre-training and this distinction between the cyclic orders is made only during model editing.

The **distance** of an edit (shown in Fig. 3) is defined as the shortest distance between the original and edited entity in the cyclic order.

### 3.3 EXPERIMENTAL SETUP

**Data-generating process.** We generate a sequence of alternating entities and relations resembling $x_1 \vec{r}_1 x_2 \vec{r}_2 x_3 \vec{r}_3 \dots$, where any consecutive triplet of entity, relation, and entity $x_i r_i x_{i+1}$ from the sequence is a fact $(x_i, \vec{r}_i, x_{i+1})$ in the knowledge graph. The composition of relations $\vec{r}_i = r_{i1} r_{i2} r_{i3} \dots$ is a sequence of 1 or more relation tokens, while $x_i$ is a single entity token. Every token is sampled using a uniform probability over all the permissible choices (see Alg. 1). For example, a plausible sequence for the example in Fig. 3 is "1 I_C4 4 III_A2 8 III_A3 3 II_C2 7", which is an alternating sequence of entities and k-hop relations. As previously noted, relations belonging to all three cyclic orders are included in the data generation process; the distinction between edit, retain, and test relations is only relevant to knowledge editing on a trained model. Furthermore, we remark that this sampling process is identical to traversing random walks on the knowledge graph, similar to previous works (Prystawski et al., 2024; Khona et al., 2024). Additional details of the generation process are documented in Appx. B.

**Training setup.** We train a Transformer model using next-token prediction on the synthetic data generated from the above data generation process. For all experiments (unless stated otherwise), we use a 2-layer nanoGPT Transformer (Karpathy, 2021). For additional details, see Appx. C.

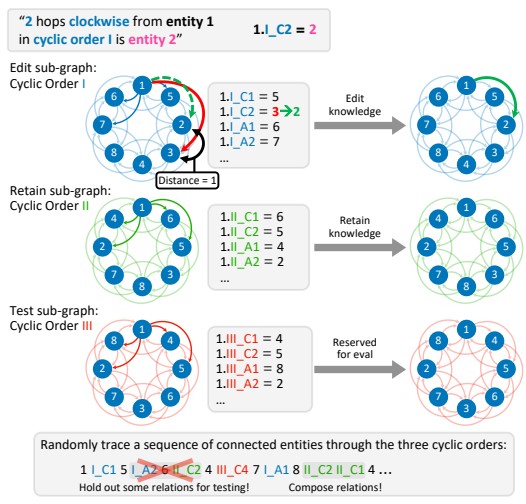

Figure 3: **Synthetic data generation process with a cyclic knowledge graph.** The entities (nodes) are arranged according to 3 different cyclic orders. Each entity (node) has relations (directed edges) pointing to 8 other entities in each cyclic order which totals to 24 relations across all 3 orders. The relations correspond to 1-4 hop neighbors in the clockwise and anti-clockwise directions. We select a random path on the knowledge graph using all 24 relations to generate a prompt, which is shown above. The relations follow the naming convention of ⟨cyclic-order⟩_⟨direction⟩⟨hops⟩, i.e. II_A3 is the relation corresponding to the three-hop anti-clockwise neighbor in the second cyclic order. In cyclic order I, the above figure denotes an edit for a relation between Entity 1 to Entity 3 (red) to a relation between Entity 1 to Entity 2 (green). The distance of the edit is 1, as defined with respect to cyclic order I.

**Evaluation (seen facts).** We assess the model's ability to remember facts seen during training, both before and after an edit. Specifically, to analyze whether the model has learned the fact $(x_i, \vec{r}, x_j)$, we prompt it with an entity $x_i$ and a relation $\vec{r}$, expecting it to produce $x_j$ as the next token. In practice, the model outputs can vary across prompts: we account for this by averaging the softmax probabilities across 5 randomly sampled sequences of the form $\dots x_i \vec{r}$ and using the output token with the highest probability.

**Evaluation (unseen facts).** We also evaluate the model on two other criteria. (1) **Compositional inference.** In addition to facts seen in the training data, we evaluate the model on compositions of relations. The model must preserve geometric structures of the data in order to compositionally generalize after a knowledge edit. (2) **Logical inference.** A key feature of reasoning in natural language is logical inference. For example, if Alice is said to be the advisor of Bob, then Bob is an advisee of Alice (even if it is not explicitly stated). Our data generation process has similar relations, such as clockwise and anti-clockwise 1-hop neighbors. By "holding out" one direction for some such pairs of relations from being observed verbatim in the training dataset, i.e., the relation may only appear compositionally, we can assess the degree to which the model internalizes properties among related relations. We can also evaluate if editing a fact for a relation changes the fact for other related relations, i.e., we check if the model's knowledge is logically self-consistent after an edit.

| Test type | (a) Unedited model | | (b) Corrective edits | | (c) $\langle\Delta\text{Acc.}\rangle$ for Counterfactual edits | | | |
|---|---|---|---|---|---|---|---|---|
| | Cyclic Order | Acc. | Sub-Graph | $\langle\Delta\text{Acc.}\rangle$ | $d=1$ | $d=2$ | $d=3$ | $d=4$ |
| Direct recall | I | 98.34 | Edit | -21.95 | -01.49 | -67.01 | -77.07 | -77.94 |
| | II | 93.71 | Retain | -22.64 | -01.91 | -66.70 | -75.49 | -75.42 |
| | III | 99.37 | Test | -21.83 | -01.75 | -67.00 | -76.12 | -77.90 |
| Logical inference | I | 98.16 | Edit | -22.24 | -01.44 | -67.22 | -77.14 | -78.02 |
| | II | 93.95 | Retain | -22.50 | -01.83 | -66.88 | -75.67 | -75.67 |
| | III | 99.40 | Test | -22.03 | -01.80 | -67.31 | -76.27 | -78.23 |
| Compositional inference | I | 88.15 | Edit | -29.60 | -05.32 | -73.15 | -80.35 | -80.63 |
| | II | 79.31 | Retain | -31.92 | -05.32 | -71.21 | -78.70 | -78.87 |
| | III | 93.50 | Test | -31.70 | -06.69 | -74.88 | -81.38 | -80.62 |

Table 1: **The direct recall, logical inference, and compositional inference accuracies before and after KE.** Results are for ROME; see Appx. F.5.1 for other methods. *(a)* The performance of our model (before editing) across the three cyclic orders (I, II, and III). Not only does our model perform well on direct recall, but it also generalizes to both logical and compositional inference tasks. This suggests that the model's internal representations extend beyond simple memorization and capture the underlying global structure that relates entities. *(b)* Changes in model accuracy after applying corrective knowledge edits. Each $\langle\Delta\text{Acc.}\rangle$ result is averaged across multiple edits, and each row labeled edit/retain/test is averaged across each of the cyclic orders *taking turns*, i.e., playing the roles of the edit, retain, and test sub-graphs. We find that corrective knowledge edits negatively affect the model's accuracy both on related and unrelated facts. These results align with the findings on LLMs (Gu et al., 2024; Gupta et al., 2024). *(c)* $\langle\Delta\text{Acc.}\rangle$ for edit, retain, and test sub-graphs after applying counterfactual edits. Intentionally introducing inconsistencies into the model's knowledge via counterfactual KE can significantly degrade model capabilities. Furthermore, the greater the induced inconsistency (scaling the counterfactual edit distance $d$ from 1-4), the more severe the resulting performance degradation.

### 3.4 REPRESENTATION SHATTERING

In this work, we explore the hypothesis that knowledge editing methods distort the geometry of the representations of entities in the knowledge graph. We believe this distortion can give us insight into why knowledge editing degrades the general capabilities of the model. In the following sections, we investigate the following hypothesis.

**Hypothesis 3.3 (Representation shattering).** *Language models embed related entities on a manifold in their internal representations. KE methods distort this manifold in order to insert new facts or alter old ones, i.e., they shatter model representations. The extent of representation shattering increases with the distance between the old fact and the desired new fact on the manifold.*

To quantify the extent of representation shattering, we define a precise metric to capture the amount of distortion of the representations:

$$R(D_*) = \frac{||D_* - D_\varnothing||_F}{||D_\varnothing||_F}, \qquad (1)$$

where $||D||_F$ is the Frobenius norm of $D$, $D_\varnothing$ the pairwise distance matrix of the entities computed using the unedited model, and $D_*$ is the pairwise distance matrix computed using the edited model. The distance between entities is computed by measuring the euclidean distance between the representation vector of each entity.

## 4 UNCOVERING REPRESENTATION SHATTERING

We study knowledge editing methods like ROME (Meng et al., 2022a), MEMIT (Meng et al., 2022b), PMET (Li et al., 2024), and AlphaEdit (Fang et al., 2024) in this work. While in the main paper we primarily present results with ROME (see Appx. C for a short primer), we provide results with other methods in Appx. F.5.1 and Appx. F.5.2. We perform two different types of edits: corrective edits and counterfactual edits. **Corrective edits** are applied to facts which the model recalls *incorrectly* after training. A **counterfactual edit** introduces a new fact, i.e., it changes fact $(x_i, r, x_j)$ to fact $(x_i, r, x_k)$ where $x_j \neq x_k$. Such an edit introduces inconsistencies in the knowledge graph.

Overall, we show the following. (1) Transformers trained on knowledge graphs recall facts, perform logical inferences, and compositional inferences. However, *both corrective and counterfactual edits*

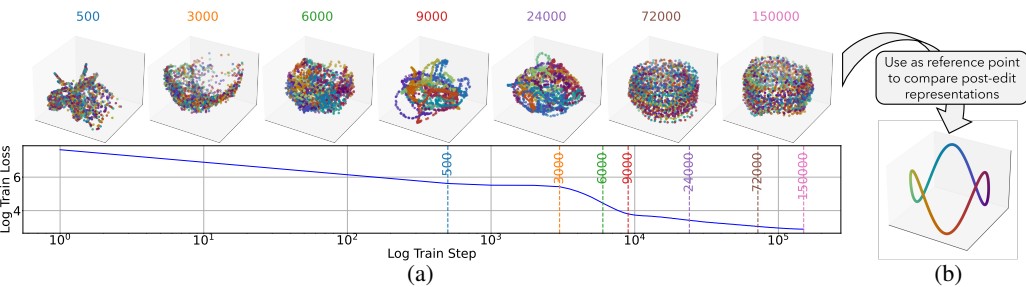

Figure 4: **Transformers learn representations that mirror the geometry of the underlying data.** *(a)* The representations—or output of the second attention layer—for the input $x\vec{r}$ for different entities $x$ and fixed relation $\vec{r}$ are visualized using Isomap. The model learns the cyclic ordering to represent all the facts. For visualizations of representations at various other model layers and modules, please see Appx. F.3. *(b)* To improve the visual fidelity of the projected representations when comparing representations of post-edit models to the unedited model, we construct Isomap neighborhood graphs using the outputs of the Transformer. For more details, please see Appx. E.

*degrade the model on all three fronts*. (2) Transformers learn a representation that reflects the underlying geometry of the data. *Knowledge edits "shatter" this representation, which serves as an explanation for the degradation in accuracy after KE.* (3) Counterfactual edits with larger distance display a larger degree of shattering. (4) *These phenomena occur in pretrained language models*, indicating representation shattering can explain degradation in model abilities after KE.

## 4.1 EVALUATING THE EFFECTS OF KNOWLEDGE EDITING

We evaluate the effects of counterfactual and corrective edits on three fronts. **Direct recall accuracy** calculates the accuracy of facts seen during training. **Logical inference accuracy** measures the accuracy on a subset of held out relations that can be inferred from other relations, i.e., the k-hop anti-clockwise neighbors can be inferred directly from the k-hop clockwise neighbors. **Compositional inference accuracy** measures the accuracy on a held out subset of compositions of two relations. Both logical inference and compositional inference measure the accuracy on samples that would be considered *out-of-distribution*.

We report scores for all three metrics in Tab. 1. The model's logical and compositional inference accuracies are close to the direct recall accuracy, which implies that the model generalizes outside of the training data before KE. **However, after KE, all accuracies decrease, with a more severe decrease for counterfactual edits** (they introduce inconsistencies between facts).

## 4.2 TRANSFORMER REPRESENTATIONS CAPTURE THE GEOMETRY OF THE DATA

The model achieves high compositional and logical inference accuracies before knowledge editing, indicating that it captures the global structure of the data and does not merely memorize all the facts seen during training. We see this reflected in the internal representation of the model (output of the second attention layer), which we visualize using Isomap (Tenenbaum et al., 2000)—a non-linear dimensionality reduction method that uses multi-dimensional scaling with distances computed using a local neighborhood graph. In Fig. 4a, we plot the evolution of the Isomap embedding—of the internal representation for the input with one entity and relation ($x\vec{r}$)—over the course of training. The different data points correspond to different values of the entity $x$, for a fixed relation $\vec{r}$ and the points in the plot are colored by the cyclic ordering. **We see that the representation manifold resembles the cyclic ordering of the entities, particularly towards the end of training.**

## 4.3 CORRECTIVE KNOWLEDGE EDITS SHATTER THE REPRESENTATION GEOMETRY

We assess how the representation changes after applying a corrective knowledge edit—i.e., applying KE to a fact that the model learned incorrectly during training. While one would expect the performance of the model to increase after a corrective edit, we find the opposite: *a corrective edit results*

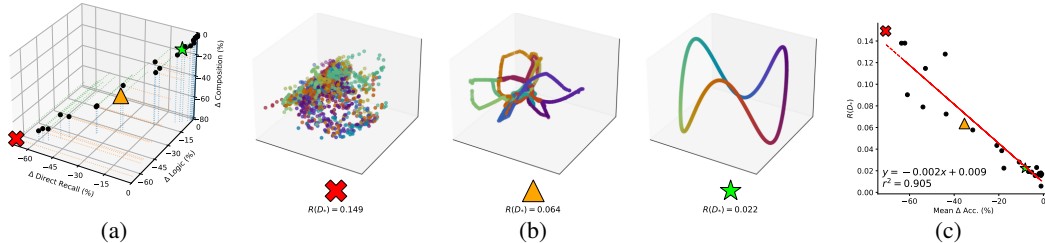

Figure 5: **Representation shattering strongly correlates with a degradation in accuracy.** *(a)* We plot the change in direct recall, logical inference, and compositional inference accuracies for different edits models, edited on different facts. We find all 3 accuracies to be strongly correlated. We select 3 edited models that span the range of accuracies, which is denoted by ✖, ▲ and ★ in the plot. *(b)* We plot the representations using a variant of Isomap (see Appx. E) with the entities colored by the cyclic order. We observe a clear trend where larger drop in accuracy directly correlates with a greater degree of representation shattering, i.e., the geometric structure of the data is destroyed after the edit. *(c)* We plot the mean drop in accuracy against the representation shattering metric $R(D_*)$ as defined in Eq. 1. Greater representation shattering is strongly correlated with more severe accuracy degradation ($r^2 = 0.905$).

*in a drop in all accuracies* (see Tab. 1). These results align with previous empirical findings showing that reasoning capabilities degrade after corrective edits (Gu et al., 2024; Cohen et al., 2023).

We visualize the representations of 3 different models using the techniques described in 4.2. The 3 models are obtained after applying 3 different edits and are selected to have high (★), intermediate (▲), and low (✖) direct recall accuracies. In Fig. 5, we observe that the model with the highest accuracy (★) has a representation that preserves the geometry of the data after the edit. However, as the model accuracy decreases, the representations also display a greater degree of distortion, no longer capturing the geometry of the data; in other words, the model is affected by representation shattering. Beyond visual inspection, this trend is also quantified in Fig. 5c, which shows a strong negative relationship between the distortion metric $R(D_*)$ (Eq. 1) and model accuracy ($r^2 = 0.905$).

### 4.4 HOW DO DIFFERENT COUNTERFACTUAL EDITS CHANGE THE EXTENT OF SHATTERING?

Counterfactual editing, wherein ones adds new facts that were unseen during training, is commonly used for evaluating KE protocols (Meng et al., 2022a; 2023; Gupta et al., 2024; Hoelscher-Obermaier et al., 2023). We consider 25 different counterfactual edits corresponding to every single counterfactual edit distance, where the counterfactual edit distance (or CE distance) is the distance between the entity in the old fact and new fact as measured in the cyclic order. Fig. 3 illustrates an example

| Sub-Graph ‖ | $d = 1$ | $d = 2$ | $d = 3$ | $d = 4$ |
|---|---|---|---|---|
| Edit | 01.80 | 21.93 | 26.22 | 27.90 |
| Retain | 01.80 | 20.84 | 25.32 | 27.28 |
| Test | 01.84 | 21.89 | 26.52 | 28.68 |

Table 2: Mean $R(D_*)$ for counterfactual edits, averaged across each sub-graph type. We observe higher degrees of representation shattering for greater counterfactual edit distances ($d$). Results are for ROME; other methods also reproduce this relationship (Appx. F.5.2).

where the counterfactual edit has an edit distance of 1. In Fig. 6, **we see that increasing the distance of the counterfactual edit results in a drop in accuracy and an increasing in the extent of shattering**. This relationship is numerically supported by $R(D_*)$ as shown in Tab. 2: shattering increases as counterfactual edit distance increases. In other words, when a new fact changes one entity to another, the extent of shattering increases as the distance between the old and new entity increases. As a naturalistic parallel, if the entities are different months, accuracy is higher when we edit "December" to "November" as opposed to "July".

### 4.5 REPRESENTATION SHATTERING IN LLMS

Finally, we investigate whether our findings generalize to large Transformers trained on naturalistic data. We consider concepts with a cyclic order, in particular months of the year, and apply a counterfactual edit to GPT-2 (Radford et al., 2019) and Mamba S4 (Gu & Dao, 2023) (see Appx. F.5.3) using

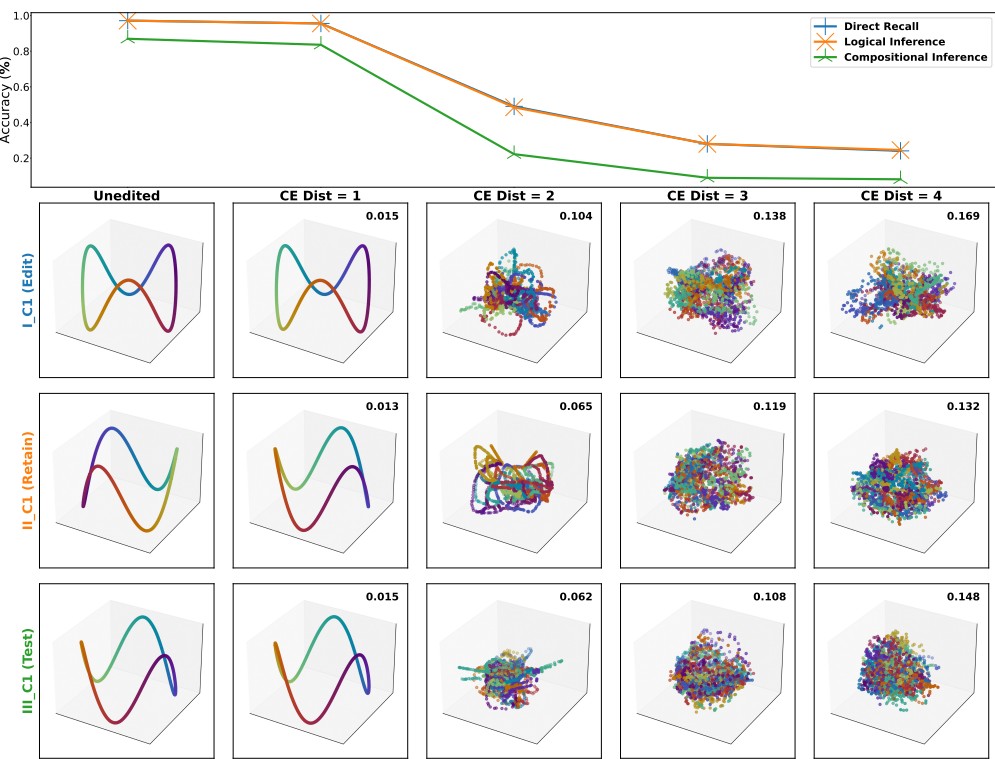

Figure 6: **Counterfactual edits with larger edit distance result in larger drop in accuracy and greater degree of representation shattering.** We apply counterfactual knowledge edits to overwrite a correctly learned fact (`1154.I_C1=567`) with inconsistent counterfactual associations. We then plot the accuracy after the counterfactual edit for different edit distances and the corresponding low-dimensional embedding of the representation obtained using a modified version of Isomap (see Appx. E). The numerical quantity in the upper right of each manifold visualization is the $R(D_*)$ value measuring the degree of representation shattering with respect to the manifold of the unedited model. Both visually and numerically, we find that a counterfactual edit with larger edit distance requires a significant distortion to the representation geometry to learn the new fact.

ROME to change the order of months. We additionally explore non-cyclic geometries, specifically tree-structured concepts, and their representation shattering in Appx. F.6.

We generated prompts following the template described in Engels et al. (2024), which include prompts such as "Let's do some calendar math. One month after January is February...". For a distance-1 edit, we modified the answer to "March"; for a distance-2 edit, we changed it to "April", and so on. We then updated the parameters of GPT-2 with these new prompt-answer pairs using ROME. Fig. 8 shows the latent representations for the 12 months extracted from the GPT-2 model before and after the edit. *We find that as we vary the edit distance from 1 to 5, the observed representation shattering increases.* In Fig. 7, we examine the impact of representation shattering on model performance. We evaluated the GPT-2 model on the reasoning task from Gu et al. (2024) both before and after editing. As the edit distance increases, we observe a gradual

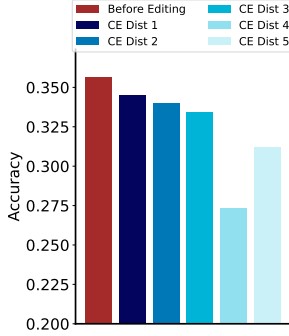

Figure 7: **Inducing representation shattering via KE degrades model performance in a real LLM (GPT-2).** We evaluate GPT-2's ability to perform a reasoning task from Gu et al. (2024) before and after editing. As the edit distance grows, accuracy gradually decreases, with a notable drop at distance 4, coinciding with the point of representation shattering.

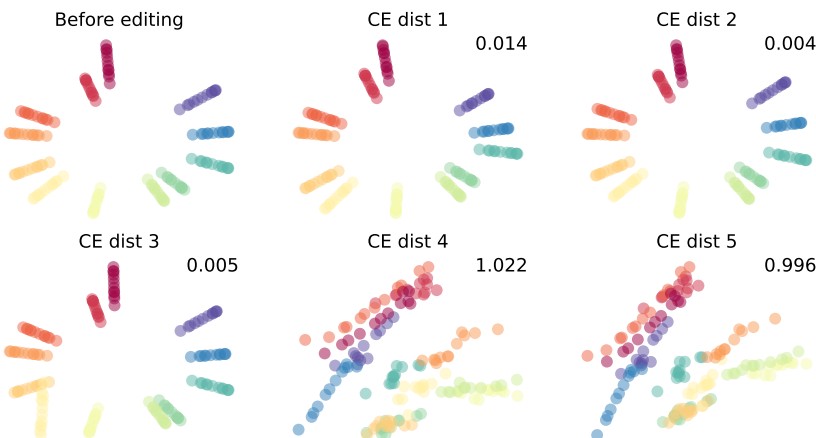

Figure 8: **Representation shattering also occurs in real LLMs (GPT-2) for months of year.** We applied KE (ROME) with counterfactual prompts for the order of months to GPT-2. The ring structure holds for edit distances up to 3 but becomes untied for distances 4 and 5. See Appx. F.5.3 for similar results with Mamba S4 (Gu & Dao, 2023), and Appx. F.6 for experiments with concepts organized in a non-circular geometry.

decline in accuracy, with a drop at distance of 4, which corresponds to the point of representation shattering. This result demonstrates that our findings from synthetic data can generalize to larger models trained on naturalistic data.

## 5 CONCLUSION

In this work, we introduced a synthetic framework to analyze the side effects of knowledge editing in transformers, identifying "representation shattering" as a key factor behind performance degradation. Specifically, we show preserving representational structures underlying a model's knowledge is crucial to avoiding negative consequences of knowledge editing: distortion of such structures impacts a model's broader capabilities. To arrive at this hypothesis, we design a controlled framework that allows investigations into models modified by knowledge editing protocols, offering clear representation-level explanations for why knowledge editing can harms models' broader capabilities that generalize to real-world models like GPT-2. While the use of simplified tasks and models can limit the scope of our conclusions, since larger, more complex real-world models may exhibit additional dynamics that our framework does not capture, we do believe that testing knowledge editing protocols on setups similar to our synthetic, knowledge graph one will significantly aid design of better editing protocols. We claim failing even such simple, albeit systematically defined settings, likely implies the editing protocol should not be readily trusted or applied at scale.

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

APPENDIX

# A    SETUP DETAILS

We will publicly release the source code for our work on GitHub at a later time.

## A.1    PSEUDO-CODE

Let $U(.)$ define the uniform distribution over the input. Let $X$ be the set of entities, $R$ the set of relations and $F$ the set of facts, defining a knowledge graph $G = (X, R, F)$.

---

**Algorithm 1:** Generate a single sequence containing a collection of facts.

```
1 function generateSequence()
2     x_p ~ U(X) from a uniform distribution over the entities.
3     S = [x_p]
4     entity_flag ← False
      // Create a sequence of alternating entities and relations
5     while len(S) < context_size do
6         if (entity_flag) then
              // Add an entity that completes a valid fact
7             Set x_n such that (x_p, r⃗, x_n) is a fact in the knowledge graph G.
8             S.append(x_n)
9             x_p ← x_n
10        else
              // Add a composition of relations
11            K ~ U({1, 2}). r⃗ = [ ]
12            for (i in 1 to K) do
13                r ~ U(R)
14                S.append(r)
15                r⃗.append(r)
16            Set x_n such that (x_s, r⃗, x_n) is a fact in the knowledge graph G.
17        entity_flag ← ¬(entity_flag)
18    S = S[:context_size]
19    return S
```

---

# B    DATA GENERATION PROCESS DETAILS

For this study, we use the following hyperparameters for our data generation process.

- **Number of entities**: 2048
- **Number of example sequences**: $10^8$
- **Maximum composition length**: 2
- **Maximum entities per sequence**: 8

Additionally, only when generating the training dataset, we drop sequences which contain one direction of a pair of conjugate facts with fixed probability $p$. In other words, if the fact $(x_i, r, x_j)$ always implies that $(x_j, r', x_i)$ is a valid fact (i.e. $r = \text{I\_C1}$ and $r' = \text{I\_A1}$), one of $(x_i, r, x_j)$ or $(x_j, r', x_i)$ may be restricted to inclusion in the training dataset by composition only (with probability $p$). Holding out these relations allows us to benchmark the model's logical inference capabilities on relations it could not have directly memorized from the training dataset. In practice, we set the probability $p = \frac{2}{3}$.

## C  Model Architecture

Our Transformer model is a fork of the open-source nanoGPT repository (https://github.com/karpathy/nanoGPT). The design is inspired by GPT, and the architecture is a decode-only Transformer with a causal self-attention mask. Our hyperparameter values are as follows.

- **Batch size**: 256
- **Context length**: 16
- **Optimizer**: Adam
- **Learning rate**: $6 \cdot 10^{-4}$
- **Training epochs**: $1.5 \cdot 10^5$
- **Decay iterations**: $1.5 \cdot 10^5$
- **Momentum**: $\beta_1 = 0.9$, $\beta_2 = 0.95$
- **Activation function**: GeLU
- **Block size**: 16
- **Embedding dimensions**: 24
- **Heads**: 12

As for tokenization, we assign every entity and relation a unique token and use standard next-token prediction with cross-entropy loss. $\text{target}_n$ is the 1-shifted version of the training sequence accounting for the padding token, and $\mathbf{x}_n$ are the logit outputs of the model at the $n$th timestep.

$$
\mathcal{L}(\mathbf{x}_n, \text{target } n) = -\log\left(\frac{\exp(\beta x_{n,\,\text{target } n})}{\sum_{v=0}^{\#\text{tokens}} \exp(\beta x_{n,v})}\right) = -\log\left(\underbrace{\text{softmax}(\beta \mathbf{x}_n)_{\text{target } n}}_{\text{prob}(\text{target } n)}\right)
$$

## D  Rank-One Model Editing (ROME)

### D.1  Algorithm Definition

Rank-One Model Editing (ROME), proposed by Meng et al. (2022a), is a popular knowledge editing algorithm used on LLMs. Their contributions are two-fold: first, through "causal tracing," they find that early-layer MLP modules of transformer models are implicated in encoding factual associations. Second, interpreting feed-forward layers as linear associative memories encoding key-value pairs, ROME applies a rank-one update to the MLP weights.

Notationally, for a factual association $(x_i, r, x_j)$, the key is the entity $x_i$ while the value is $x_j$. In each feed-forward layer, the hidden state $\mathbf{h}_i^{(l-1)}$ at layer $l-1$ is transformed into a key $\mathbf{k}$ by the weight matrix $\mathbf{W}_{fc}^{(l)}$, and the corresponding value $\mathbf{v}$ is retrieved by the matrix $\mathbf{W}_{proj}^{(l)}$:

$$
\mathbf{h}_i^{(l)} = \mathbf{W}_{proj}^{(l)} \sigma\left(\mathbf{W}_{fc}^{(l)} \mathbf{h}_i^{(l-1)}\right)
$$

where $\sigma(\cdot)$ denotes the activation function.

To modify the factual association $(x_i, r, x_j)$ in the model, ROME computes a new key-value pair $(\mathbf{k}^*, \mathbf{v}^*)$, representing the entity $x_i$ and the new target entity $x_j^*$. ROME then applies a rank-one update to the weight matrix $\mathbf{W}_{proj}^{(l^*)}$ at a specific layer $l^*$ to encode this new fact:

$$
\hat{\mathbf{W}}_{proj}^{(l^*)} = \mathbf{W}_{proj}^{(l^*)} + \lambda\left(\mathbf{C}^{-1}\mathbf{k}^*\right)^\top \quad \text{where } \lambda = \frac{\mathbf{v}^* - \mathbf{W}_{proj}^{(l^*)}\mathbf{k}^*}{\left(\mathbf{C}^{-1}\mathbf{k}^*\right)^\top \mathbf{k}^*}
$$

Here, $\mathbf{C}$ is the uncentered covariance matrix of the key vectors $\mathbf{k}$, estimated by sampling tokens from a representative dataset.

The key vector $\mathbf{k}^*$ corresponds to the entity $x_i$ in the factual association $(x_i, r, x_j^*)$. The vector is computed by averaging the MLP output for $x_i$ over multiple randomly generated contexts:

$$\mathbf{k}^* = \frac{1}{N} \sum_{j=1}^{N} \sigma \left( \mathbf{W}_{fc}^{(l^*)} \gamma \left( \mathbf{a}_i^{(l^*)} + \mathbf{h}_i^{(l-1)} \right) \right)$$

where $\gamma(\cdot)$ is a normalization function, and $\mathbf{a}_i^{(l^*)}$ is the attention output at layer $l^*$.

The value vector $\mathbf{v}^*$ is optimized to maximize the model's probability of predicting the target entity $x_j^*$ given the subject $x_i$ and relation $r$. This is done by minimizing the following objective:

$$L(\mathbf{z}) = \frac{1}{N} \sum_{j=1}^{N} \left( -\log P \left( x_j^* | x_i, r \right) + D_{KL} \left( P_G \left( x_i | p' \right) || P_G \left( x_i | p' \right) \right) \right)$$

The first term maximizes the probability of the target entity $x_j^*$, while the second term controls for "essence drift" to retain information about $x_i$. This is done by sampling inputs $p'$ for which the model's outputs should not change during the edit.

## D.2 IMPLEMENTATION

In our implementation of ROME tailored to our model, we apply the edit at layer 1 as it is the only available early-site layer in our model configuration. The covariance matrix $\mathbf{C}$ is estimated by randomly sampling $10^5$ inputs from the validation dataset. This provides a representative set of key vectors for computing the rank-one update. To solve for the key vector $\mathbf{k}^*$, we sample $10^5$ random context sequences, with sequence lengths varying between 2 and 10 tokens. The value solver follows a similar procedure by sampling $10^2$ context sequences selected in the same manner as the key solver. The value optimization is performed using the Adam optimizer, with hyperparameters lr $= 10^{-3}$ and weight decay $= 10^{-4}$. The value solver optimizes between 5 and 500 iterations, stopping when the predicted token is replaced by $x_j^*$. The KL divergence weight is set to 3 during optimization.

# E  VISUALIZATION METHODS

In Fig. 4, we demonstrated the emergence of cyclic representations within the model by extracting representations and generating 3D Isomap projections. While the visualizations support the notion that cyclical representations are present in the model, changes in the projections can be difficult to intuitively interpret due to the overlap of differently colored segments of the manifold. For example, below is a recreation of Fig. 6 using raw Isomap projections.

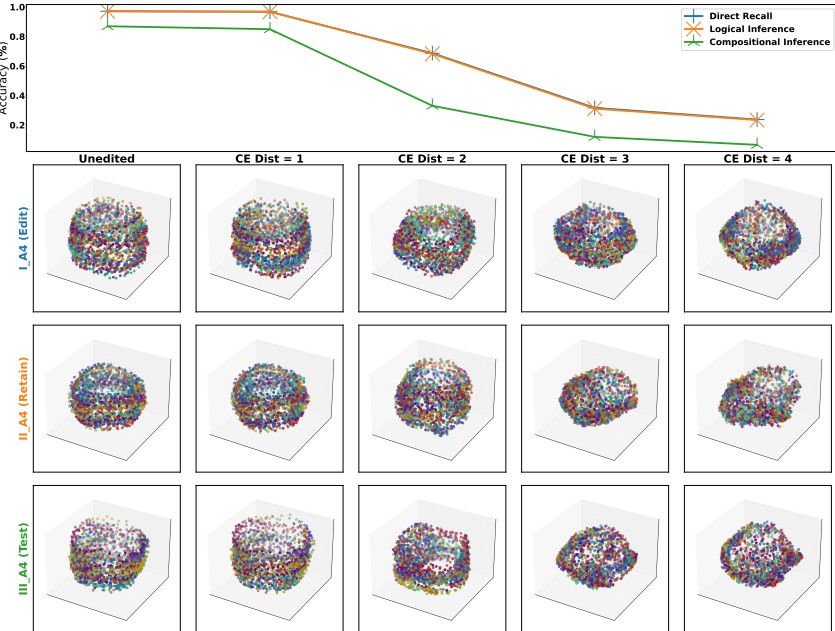

Figure 9: An equivalent version of Fig. 6 using the unprocessed Isomap projection renderings. Representation shattering is still visible in the flattening and clustering of points in the manifold as the counterfactual edit distance increases.

The coinciding ring segments are an artifact of the lossy projection of high-dimensional cyclical representations into a low-dimensional space: when dimensionality reduction to 3D is applied, the high-dimensional cyclical structure gets "squished" into a torus. To enhance the visual perceptibility of the representation shattering phenomenon, we additionally implement a pre-processing step to constrain the construction of the Isomap neighbors graph using the model's output predictions. More concretely, when visualizing the post-edit manifold for a particular edit $(x_i, r, x_j^*)$, we adopt the following procedure:

1. Construct a set $S_0$ of entities by prompting the *unedited* model for all immediate neighbors of $x_i$ in the cycle order of $r$ (i.e. by getting outputs for $x_i r'$ for all $r'$ in the same cycle order as $r$).
2. Apply the knowledge edit.
3. Construct a set $S_1$ of entities by collecting outputs from the *edited* model for all $s_i r$ where $s_i \in S_0$.
4. Constrain the Isomap pair-wise distance matrix to members of $S_1$.

This procedure remains faithful in comparing the pre-edit model to the post-edit model, as relies solely on model predictions and does not introduce any ground-truth priors.

# F ADDITIONAL RESULTS

## F.1 INDEPENDENCE OF SUBGRAPHS

In our evaluations, we make edits to various relations under the assumption that the Transformer internalizes the independence of the cyclic orders (`I`, `II`, and `III`). Here, we ask: do the model's internal representations truly reflect this? We answer this question by inspecting the representations for the output of the multi-head attention output in layer 2 at the last token position using PCA. Unlike in previous sections where we focused on a fixed relation $r$ and varied $x_i$ for inputs of the form $\cdots x_i r$, we now vary both $x_i$ and $r$ and color-code each projection by the cyclic order to which the relation $r$ belongs. We present the resulting projections in Fig. 10, and find that prompts eliciting knowledge for each cyclic order are clustered closely together in the latent space—this is further evidence that the model internalizes the properties of the underlying knowledge graph.

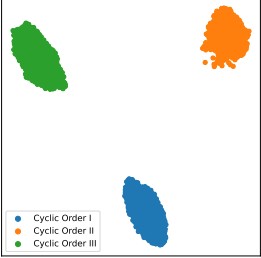

Figure 10: PCA of representations extracted from the output of the multi-head attention output in layer 2 at the last token position, color-coded by the cyclic order of the last relation token.

## F.2 MANIFOLDS FOR ALL RELATIONS

In Fig. 11, we provide isomap projections of representations extracted for all relations from our model. We show highly structured representations are formed within the model, indicating the model is truly learning the data-generating process and not merely memorizing information.

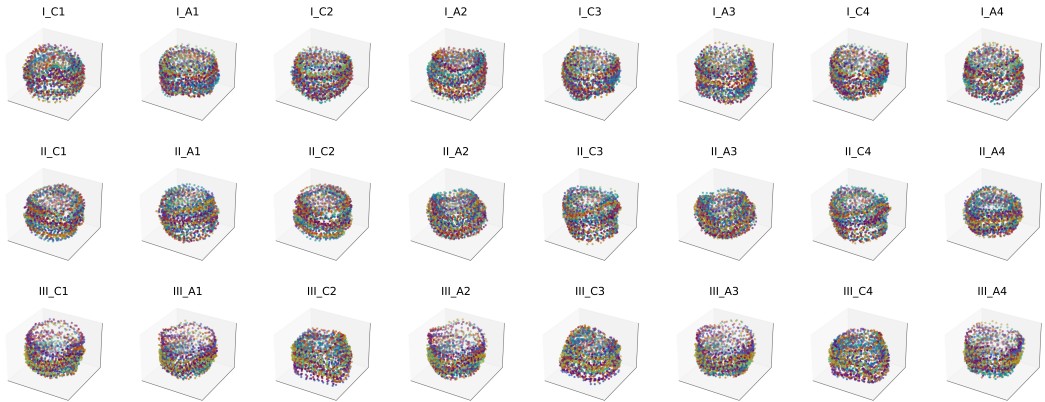

Figure 11: Isomap projections for representations for all relations, extracted from the output of the multi-head attention output in layer 2 at the last token position. We find that all relations are represented by a cyclical representation manifold. This shows that the model is not falling back on memorization for any relations—rather, it represents all of its knowledge in consistent, ring-like manifolds.

### F.3 MANIFOLDS FOR VARIOUS REPRESENTATION EXTRACTION POINTS

We repeat our representation visualizations analysis for all relations at different layers in the model and at different sequence positions, finding the structured representations are found at specific token positions. See Fig. 12.

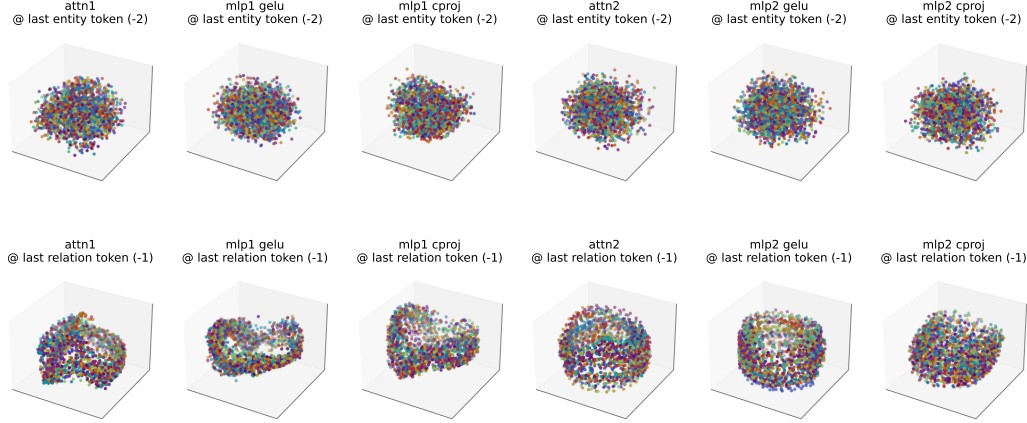

Figure 12: 3D Isomap projections for representations extracted from various points in the model for various token positions. The cyclical representation manifolds can only be observed for the last relation token position (−1th token), and not at the last entity token position (−2th token). This intuitively makes sense because the last relation token informs the model about which cycle order the current input is querying for. We primarily use the "attn2 last relation token" representations throughout this work because it is the earliest point at which a well-structured cyclical manifold can be observed beyond the point of the ROME intervention (which is at "mlp1 cproj").

## F.4 COUNTERFACTUAL EDITING

### F.4.1 DISTRIBUTION OF DEGREDATIONS FOR COUNTERFACTUAL EDITS

The plots in Fig. 13 correspond to the counterfactual editing results presented in Sec. 4.4 and Tab. 1.

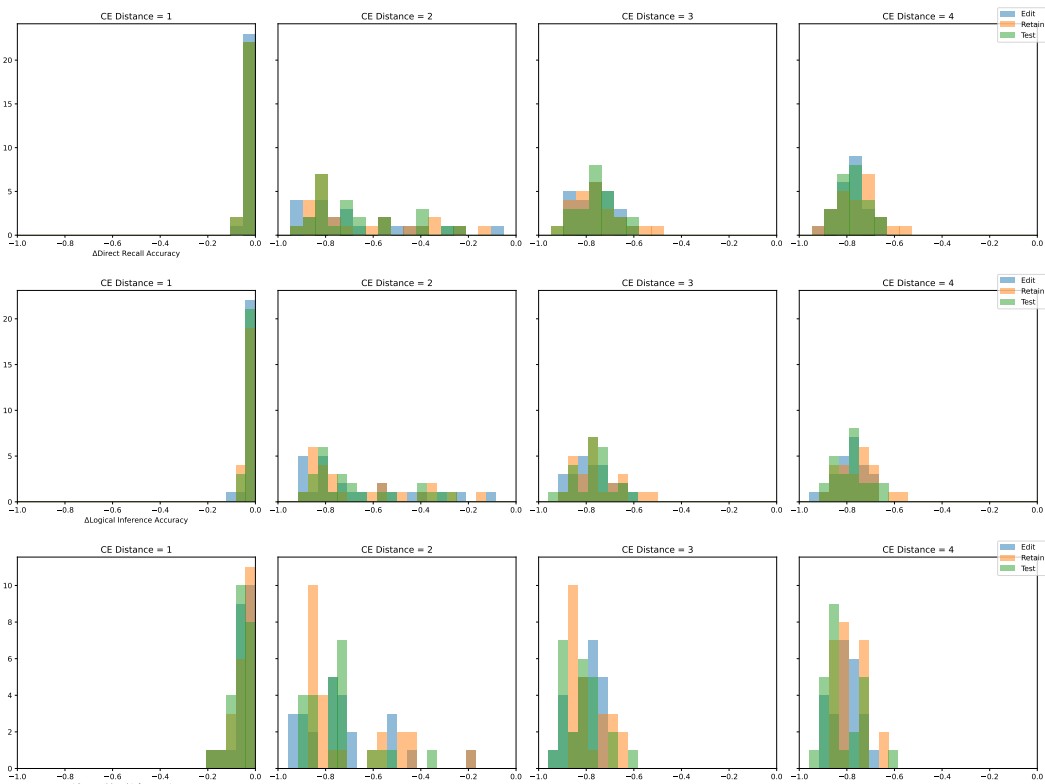

Figure 13: Distribution of post-edit accuracy degredations for direct recall, logical inference, and compositional inference in relation to the counterfactual edit distances. A significant shift can be observed between CE distances of 1 and 2, showing the point at which detrimental representation shattering can occur.

### F.4.2 ADDITIONAL VISUALIZATIONS

In Fig. 6, we showcase an example of the change in accuracies and representation manifolds when applying a counterfactual edits (specifically for fact 1154.I_C1). For a more representative view, we additionally provide more examples of counterfactual edits (with both raw and pre-processed versions side-by-side, as described in Appx. E).

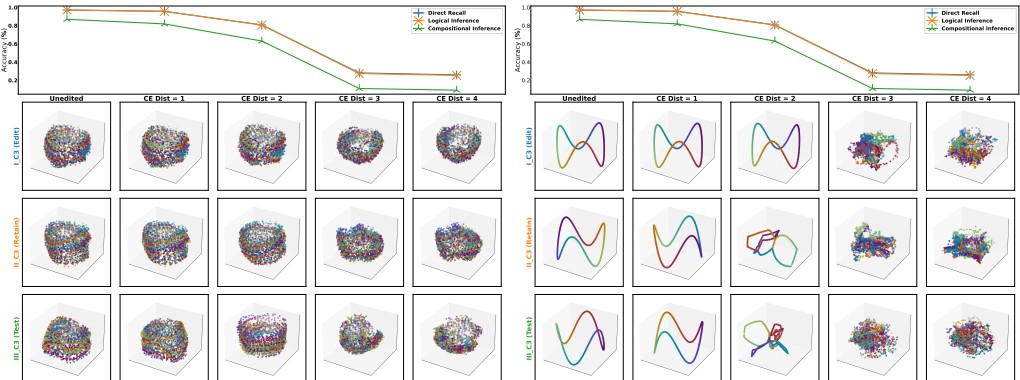

Figure 14: Counterfactual editing visualizations for 1623.I_A2.

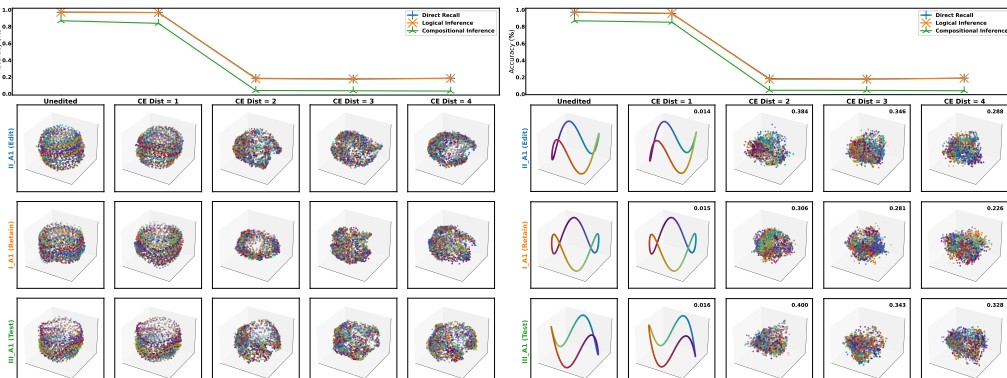

Figure 15: Counterfactual editing visualizations for 1121.II_C1.

## F.5 ALTERNATIVE EDITING METHODS AND MODELS

### F.5.1 MODEL ACCURACY

In Tab. 1, we evaluate the effects of corrective and counterfactual edits with ROME with respect to changes in the model's direct recall accuracy, logical inference accuracy, and compositional inference accuracy. The results give several key insights: corrective knowledge edits negatively affect the model's accuracy both on related and unrelated facts, intentionally introducing inconsistencies into the model's knowledge via counterfactual KE can significantly degrade model capabilities, and greater induced inconsistency (scaling the counterfactual edit distance $d$ from 1-4) causes greater performance degradation. Now, we reinforce these findings by repeating the same edits and evaluations with additional KE methods: namely **MEMIT** (Meng et al., 2023), **AlphaEdit** (Fang et al., 2024), and **PMET** (Li et al., 2024). We present our results in Tab. 3.

| KE Method | Test type | Corrective edits | | $\langle\Delta\text{Acc.}\rangle$ for Counterfactual edits | | | |
|---|---|---|---|---|---|---|---|
| | | Sub-Graph | $\langle\Delta\text{Acc.}\rangle$ | $d=1$ | $d=2$ | $d=3$ | $d=4$ |
| ROME | Direct recall | Edit | -21.95 | -01.49 | -67.01 | -77.07 | -77.94 |
| | | Retain | -22.64 | -01.91 | -66.70 | -75.49 | -75.42 |
| | | Test | -21.83 | -01.75 | -67.00 | -76.12 | -77.90 |
| | Logical inference | Edit | -22.24 | -01.44 | -67.22 | -77.14 | -78.02 |
| | | Retain | -22.50 | -01.83 | -66.88 | -75.67 | -75.67 |
| | | Test | -22.03 | -01.80 | -67.31 | -76.27 | -78.23 |
| | Compositional inference | Edit | -29.60 | -05.32 | -73.15 | -80.35 | -80.63 |
| | | Retain | -31.92 | -05.32 | -71.21 | -78.70 | -78.87 |
| | | Test | -31.70 | -06.69 | -74.88 | -81.38 | -80.62 |
| MEMIT | Direct recall | Edit | -09.51 | -01.64 | -57.98 | -67.04 | -68.72 |
| | | Retain | -07.08 | -01.78 | -48.68 | -57.23 | -58.52 |
| | | Test | -06.54 | -01.19 | -51.85 | -63.96 | -70.26 |
| | Logical inference | Edit | -09.58 | -01.61 | -58.16 | -67.31 | -69.10 |
| | | Retain | -06.73 | -01.64 | -48.45 | -57.55 | -58.66 |
| | | Test | -06.67 | -01.37 | -52.37 | -64.65 | -70.99 |
| | Compositional inference | Edit | -11.43 | -01.85 | -57.79 | -67.82 | -71.79 |
| | | Retain | -08.34 | -00.68 | -53.05 | -62.71 | -64.09 |
| | | Test | -10.47 | -03.30 | -53.36 | -66.81 | -73.42 |
| AlphaEdit | Direct recall | Edit | -06.05 | -01.45 | -54.68 | -64.01 | -63.48 |
| | | Retain | -04.68 | -01.69 | -43.72 | -52.36 | -53.63 |
| | | Test | -03.75 | -00.92 | -47.53 | -59.57 | -66.09 |
| | Logical inference | Edit | -06.13 | -01.42 | -54.93 | -64.42 | -63.91 |
| | | Retain | -04.37 | -01.55 | -43.58 | -52.74 | -53.93 |
| | | Test | -03.85 | -01.03 | -48.05 | -60.38 | -66.83 |
| | Compositional inference | Edit | -07.75 | -01.72 | -55.82 | -66.42 | -68.35 |
| | | Retain | -05.99 | -00.08 | -50.19 | -59.62 | -61.57 |
| | | Test | -07.03 | -02.75 | -51.14 | -64.14 | -70.95 |
| PMET | Direct recall | Edit | -03.97 | -01.34 | -48.27 | -50.80 | -54.72 |
| | | Retain | -02.78 | -01.61 | -35.54 | -39.18 | -46.36 |
| | | Test | -02.01 | -00.98 | -43.40 | -44.29 | -52.67 |
| | Logical inference | Edit | -04.02 | -01.32 | -48.48 | -51.05 | -55.06 |
| | | Retain | -02.47 | -01.47 | -35.40 | -39.39 | -46.60 |
| | | Test | -02.10 | -01.11 | -44.07 | -44.76 | -53.32 |
| | Compositional inference | Edit | -05.60 | -01.37 | -49.89 | -55.65 | -60.62 |
| | | Retain | -03.09 | -00.23 | -42.24 | -47.87 | -53.78 |
| | | Test | -04.56 | -02.95 | -47.00 | -50.95 | -58.98 |

Table 3: Results of Tab. 1, replicated using MEMIT (Meng et al., 2023), AlphaEdit (Fang et al., 2024), and PMET (Li et al., 2024). Overall, recent methods succeeding ROME are slightly less damaging to model accuracy. However, all evaluated methods nonetheless cause undesirable performance degradations in similar ways to ROME (especially for increased counterfactual edit distances). This suggests that KE methods, despite their differences in approaches, often suffer from similar shortcomings in terms of negatively impacting model performance.

### F.5.2 REPRESENTATION SHATTERING METRIC

In Tab. 2, we showed that increasing the distance of the counterfactual edit results in an increase in the extent of shattering, as numerically captured by $R(D_*)$. In similar spirit to Appx. F.5.1, we seek to verify whether this relationship between counterfactual edit distance and representation shattering holds for methods other than ROME, i.e. MEMIT (Meng et al., 2023), AlphaEdit (Fang et al., 2024), and PMET (Li et al., 2024). We present our results in Tab. 4.

| Method | Sub-Graph | $d = 1$ | $d = 2$ | $d = 3$ | $d = 4$ |
|--------|-----------|---------|---------|---------|---------|
| ROME | Edit | 01.80 | 21.93 | 26.22 | 27.90 |
| | Retain | 01.80 | 20.84 | 25.32 | 27.28 |
| | Test | 01.84 | 21.89 | 26.52 | 28.68 |
| MEMIT | Edit | 01.89 | 08.58 | 09.32 | 08.78 |
| | Retain | 01.86 | 07.31 | 07.66 | 07.50 |
| | Test | 01.85 | 07.49 | 08.35 | 07.70 |
| AlphaEdit | Edit | 01.86 | 07.77 | 08.44 | 07.68 |
| | Retain | 01.85 | 06.51 | 06.89 | 06.99 |
| | Test | 01.83 | 06.89 | 07.60 | 06.99 |
| PMET | Edit | 01.83 | 06.55 | 06.44 | 06.41 |
| | Retain | 01.84 | 05.45 | 05.42 | 05.85 |
| | Test | 01.83 | 06.14 | 05.75 | 06.31 |

Table 4: Results from Tab. 2, replicated using the alternative knowledge editing methods of MEMIT (Meng et al., 2023), AlphaEdit (Fang et al., 2024), and PMET (Li et al., 2024). These successors to ROME achieve lower amounts of representation shattering overall, coinciding with their more favorable performance in Appx. F.5.1. However, the relationship between greater counterfactual edit distance $d$ and greater representation shattering $R(D_*)$ still robustly holds for all methods. This result again shows that various KE methods struggle in similar ways: specifically, the greater the inconsistency between the model's original knowledge and the edited fact, the greater the resulting distortion upon the model's representations.

### F.5.3 ROME ON MAMBA

In Sec. 4.5, we investigate whether the representation shattering hypothesis generalizes to large Transformers trained on naturalistic data. We consider the cyclic order of the months of the year and apply a counterfactual edit to GPT-2 (Radford et al., 2019) and found that as we vary the edit distance from 1 to 5, the observed representation shattering increases.

To further probe the robustness of our claims with respect to model size and model architecture, we additionally explore KE with Mamba (Gu & Dao, 2023). Mamba is a structured state space sequence model, and we use the Mamba-2.8B variant for this experiment. For consistency with previous experiments, we use ROME as the editing method, adapted appropriately to work with the Mamba architecture (Sharma et al., 2024). As for the counterfactual edit prompts, we use the same prompts as in Sec. 4.5 (i.e. "`Let's do some calendar math. One month after {}` `is {}`"). We present the resulting manifold visualizations and $R(D_*)$ values in Fig. 16.

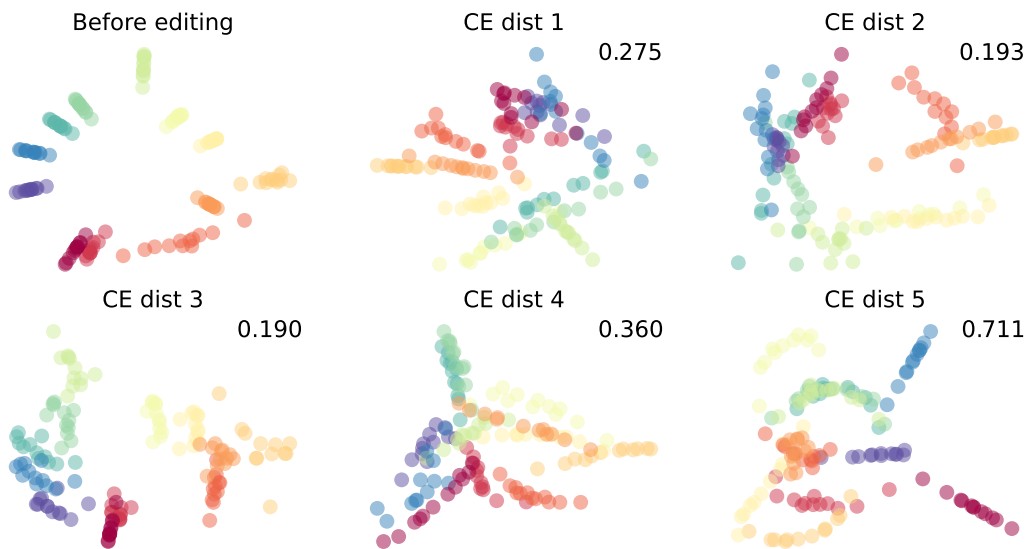

Figure 16: Fig. 8, replicated using Mamba-S4 (Gu & Dao, 2023) and ROME. Like ROME applied to GPT-2, the ring structure shatters for larger counterfactual edit distances. Interestingly, the degree of shattering fluctuates more in Mamba-S4 with respect to the counterfactual edit distance than in GPT-2. This may be caused by Mamba's greater model complexity, as its representation manifolds likely encode more information about months of the year than just their cyclic order in the calendar. Nonetheless, larger counterfactual edit distances (i.e. distance 5) causes greater shattering than smaller counterfactual edit distances (i.e. distance 1), demonstrating that our findings are not limited to GPT-2 and can be extended to other models and architectures.

## F.6 KNOWLEDGE EDITING WITH NATURALISTIC TREES

In our experiments, we primarily focus on synthetic knowledge graphs with cyclical structures. While the simplicity of cycles is desirable for our synthetic experiments, real human knowledge and language can exhibit more complex structures. For example, geographical ground-truths can be expressed in a tree structure, with entities like cities/countries/continents having relations with other cities/countries/continents, i.e. $x_i =$ Paris, $r =$ located in country, $x_j =$ France.

Here, we ask: does the representation shattering hypothesis hold for more realistic tree-shaped knowledge graphs in more complex models like GPT-2? To answer this question, we take inspiration from the classic "The Eiffel Tower is located in the city of Rome" example of counterfactual knowledge editing (Meng et al., 2022a). For our purposes, we edit the country associations of major cities. In particular, we consider the following five countries: *France*, *Spain*, *Italy*, *Germany*, and *the United Kingdom*. Then, we also consider the five most populous cities of each country, totaling 25 cities: *Paris*, *Marseille*, *Lyon*, *Toulouse*, *Nice*, *Madrid*, *Barcelona*, *Valencia*, *Sevilla*, *Zaragoza*, *Rome*, *Milan*, *Naples*, *Turin*, *Palermo*, *Berlin*, *Hamburg*, *Munich*, *Köln*, *Frankfurt am Main*, *London*, *Birmingham*, *Liverpool*, *Glasgow*, and *Sheffield*. The knowledge graph involving these city-country pairs contains facts such as ($x_i =$ Paris, $r =$ located in country, $x_j =$ France). The ground-truth arrangements of the cities and countries form a tree (Fig. 17a).

From the latent space of LLMs, however, it is difficult to extract clean tree-like geometries. When we project the representations for tokens corresponding to the country and city names using Isomap, the result does not yield a discernible tree shape (Fig. 17b). Despite the exact structure of the latent space not being clear, the notion of "distance" in the manifold can still be applied. For example, in Fig. 17b, *Spain* is closer to *France* than is *the United Kingdom*; therefore, the edit "Paris is a city in the country of Spain" has a smaller counterfactual edit distance than does the edit "Paris is a city in the country of the United Kingdom." Fig. 18a and Fig. 18b show the representation manifold Isomaps after applying the edits "Paris is a city in the country of Spain" and "Paris is a city in the country of the United Kingdom," respectively, using ROME on GPT-2. First, we find that both counterfactual edits cause the representations for all cities and countries to collapse inward. Moreover, the edit to "the United Kingdom" causes a greater distortion than the edit to "Spain," as is evident both by visual inspection and by the numerical representation shattering quantity $R(D_*)$.

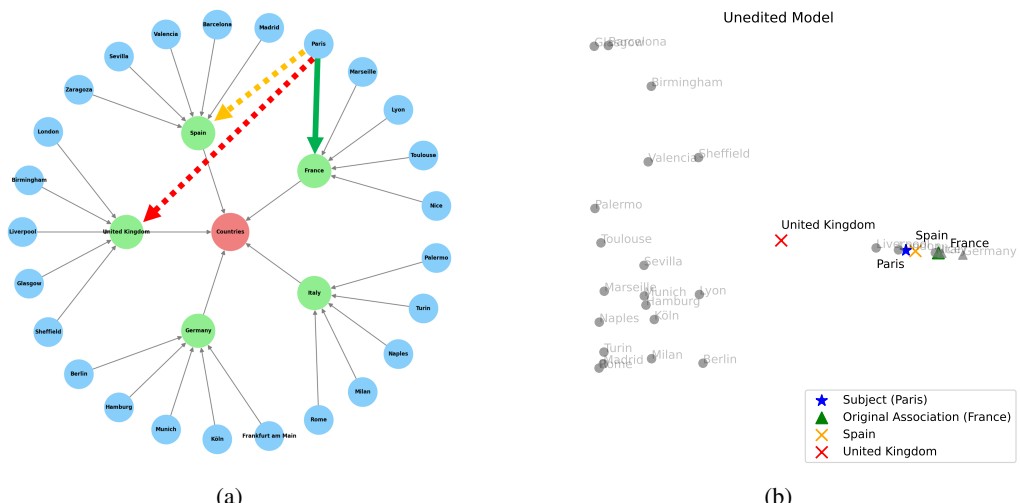

(a)                                                        (b)

Figure 17: *(a)* The ground-truth tree representing the 5 countries and its 25 cities. The correct factual association for the prompt "Paris is a city in the country of..." is France. In this example, we consider the counterfactual edits "Paris is a city in the country of Spain" and "Paris is a city in the country of the United Kingdom". *(b)* Isomap projections of representations for the selected countries and cities. We find that, on this model's representation manifold, editing Paris to be in Spain constitutes a smaller counterfactual edit distance than does editing Paris to be in the United Kingdom.

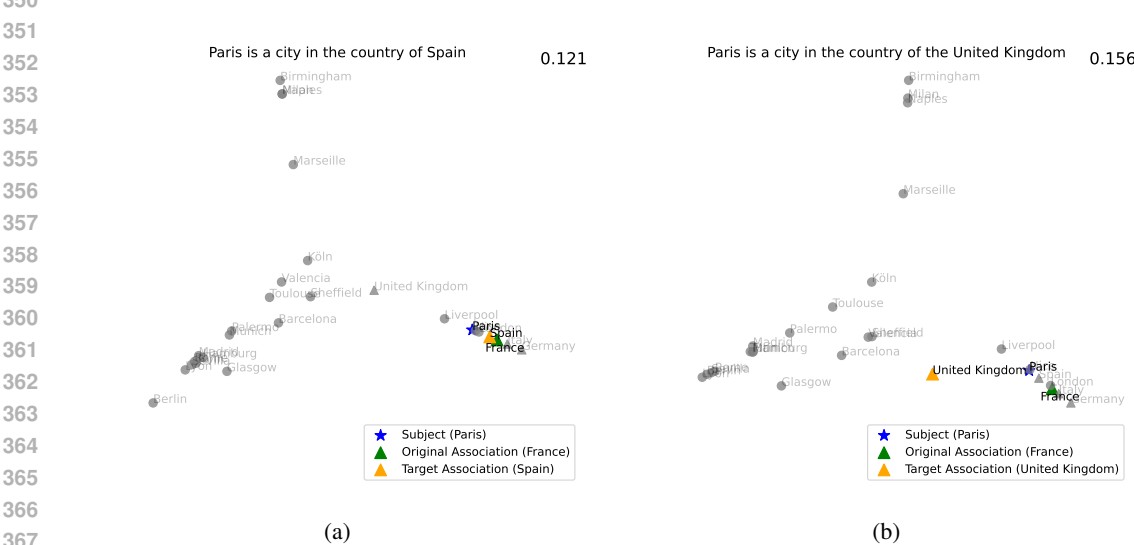

Figure 18: Isomap projections of latent representations after applying a counterfactual edit. *(a)* "Paris is a city in the country of Spain." *(b)* "Paris is a city in the country of the United Kingdom."

To take a step in verifying whether this finding is generalizable, we applied counterfactual edits to each of the 25 selected cities. For each city, we computed the country which constitutes the "closest" and "furthest" counterfactual edit distance on the model's representation manifold. After applying the two counterfactual edits, we computed $R(D_*^{\text{farthest}})$ and $R(D_*^{\text{closest}})$. Across the 25 cities, the average ratio $R(D_*^{\text{farthest}})/R(D_*^{\text{closest}})$ was $1.1483$. In other words, when changing a city's parent country, editing to a close country on the representation manifold yields less shattering than editing to a country which sits far away on the manifold.

These preliminary results align with our main hypothesis: KE methods distort language models' representations in order to insert new facts or alter old ones (i.e. representation shattering), and the extent of representation shattering increases with the distance between the old fact and the desired new fact on the manifold.

