# OpenReview forum: "Representation Shattering in Transformers: A Synthetic Study with Knowledge Editing"
_ICLR.cc/2025/Conference — Submitted to ICLR 2025_

### Official Review · Reviewer_jMeL · 2024-10-31

**Soundness:** 3
**Presentation:** 2
**Contribution:** 3
**Rating:** 5
**Confidence:** 2

**Summary:**

This paper presents a mechanistic explanation of the ripple effects of knowledge editing. This explanation considers how the representations of the model changes during the KE. This paper considers a particular cyclic structure. When the model’s training data are generated from this cyclic structure, the learned representation gradually forms a geometry that resembles the cyclic structure of the data. KE, both corrective edits and counterfactual edits, shatter this cyclic structure. In this way, the ripple effect change in the model’s behavior after edits can be illustrated by the change in the representations.

**Strengths:**

- The ripple effect of knowledge edit has been studied, but the prior works have not explained the effects from a mechanistic viewpoint. This paper takes a nice step toward explaining this effect mechanistically.
- The angle toward explaining this ripple edit effect is interesting (despite a bit niche). Cyclic structure is a commonly occurring structure in natural language, and there is a nice illustration in the representation space.
- This paper uses a nice combination of qualitative and quantitative illustrations to describe the representation shattering effect.

**Weaknesses:**

- The quantitative explanation part can take up more space in the writing. Currently only Figure 5(c) involves quantitative description of representation shattering, and other parts are qualitative. Table 1 presents the overall effects of knowledge editing, which is not really the quantitative descriptions.
- Additionally, the definition for describing the representation shattering effect should be introduced earlier. Currently it’s hidden at lines 413-415 at page 8, closer to the end of the paper. This should be moved to the front.
- Subsection 3.3 can be made clearer. Based on the example presented, I struggled at understanding how the nodes are connected, and e.g., what I/III/ABC refer to.
- The effects are tested on a 2-layer nanoGPT Transformer, plus a GPT2. These limit the generalizability of the finding. I’d recommend experimenting on at least two models on each size — perhaps containing a mixture of uni-directional models (like *GPT) and bidirectional models (like BERT).

**Questions:**

Could you elaborate on the synthetic data generation process? Do A/B/C refer to the entities and the numbers refer to the edges?

---

> ### Author Response · Authors · 2024-11-28
> **Rebuttals (1/1)**
>
> We thank the reviewer for the feedback and for taking the time to review our work. We are glad that the reviewer finds our qualitative and quantitative results take an important step towards building a mechanistic view of knowledge editing.
>
> ------
> ------
>
> ### Main comments
>
> > **I'd recommend experimenting on at least two models on each size--perhaps containing a mixture of uni-directional models (like GPT) and bidirectional models (like BERT).**
>
> Thank you for this suggestion! While we were not able to run two models of each size due to time constraints, we extended our experiments beyond GPT-2 by replicating key results with the Mamba-S4 2.8B model. The results are now added to Appendix F.5.3. Both the quantitative and qualitative metrics point to a distortion in the representation geometry, i.e., representation shattering is seen in other architectures (appendix F.5.3), as well as other knowledge editing methods (appendix F.5.2). We also note that we are open to adding further experiments with more models in the final version of the paper.
>
> ------
>
> > **The quantitative explanation part can take up more space in the writing.**
> > **The definition for describing the representation shattering effect should be introduced earlier.**
>
> We appreciate this suggestion and note that we have now added a new section (3.4) on representation shattering that more prominently defines the central hypothesis being explored, and presents a quantitative measure of representation shattering. In particular, we define $R(D_*) = ||D_* - D_\varnothing|| / ||D_\varnothing||$ as the degree of distortion, where the norm is the Frobenius norm, $D_\varnothing$ is the pairwise distances between the representations of entities of the unedited model, and $D_*$ is the pairwise distances between representations of the edited model. **We have added this metric to all our results throughout the paper.**
>
> ---
>
> > **Currently only Figure 5(c) involves quantitative description of representation shattering, and other parts are qualitative**
>
> As we remarked above, we have now added a quantitative metric to supplement the qualitative pictures in the manuscript. In particular, Figure 6 and Figure 7 now report the representation distortion measure $R(D_*)$.  Appendix F.5.2 contains an extended version of Table 1 with $R(D_*)$ reported alongside the factual recall accuracy. Across **all figures and tables**, we find that the distortion measure is consistent with the qualitative picture and is negatively correlated with the factual recall accuracy. Thank you for this suggestion!
>
> ------
>
> > **Could you elaborate on the synthetic data generation process? Do A/B/C refer to the entities and the numbers refer to the edges?... I struggled at understanding how the nodes are connected, and e.g., what I/III/ABC refer to.**
>
> Thank you for raising this point! We have added a few paragraphs (highlighted in blue) to section 3.2 to further clarify our data-generating process. We summarize the same below.
>
> We define a knowledge graph with 2048 entities (denoted by 1-2048) over which we define 3 cyclic orders (order I, II and III). The cyclic orders arrange the entities into a cycle and each cyclic order is a random permutations of the entities. We create 8 relations for each cyclic order totaling to 24 relations. The 8 relations correspond to the 1-hop, 2-hop, 3-hop and 4-hop neighbors in the clockwise and anti-clockwise directions in the cycle. The relations are named after a combination of the cyclic order (I, II, III), the neighbor’s distance (between 1-4) and the neighbor’s direction (Clockwise, Anti-clockwise). For instance, the relation “I_C2” denotes the 2-hop neighbor in the clockwise direction, with respect to cyclic order I.” The fact  (5, II_C3, 23) tells us that entity “23” is the clockwise 3-hop clockwise neighbor of entity “5”.
>
> ------
>
> > **Subsection 3.3 can be made clearer.**
>
> Thank you for the suggestion! We have rewritten this section and improved its organization as well. We would be happy to make changes to the paper if you find any particular parts of subsection 3.3 are still unclear.
>
> ----
> ----
>
> ### Summary
>
> We thank the reviewer for their feedback that has helped us substantially improve the clarity of our paper! Specifically, we have now added **a precise quantification of representation shattering** throughout our results and **reworked Section 3** to improve the clarity of our data-generation process. We have also added several new experiments that confirm the robustness of our claims across **different editing methods, model architectures, and graph structures**. We hope these updates help address the reviewer's concerns, and, if so, that they would consider raising their score to support our work's acceptance!

---

> ### Author Response · Authors · 2024-12-01
> **Discussion Period Reminder**
>
> Dear Reviewer,
>
> We thank you again for your detailed feedback on our work. Given the discussion period ends soon, we wanted to check in if our provided responses address your concerns, and see if there are any further questions that we can help address.
>
> Thank you!

---

> ### Author Response · Authors · 2024-12-02
> **Follow-up Reminder**
>
> Dear reviewer,
>
> This is a gentle reminder that the discussion period is ending soon. We have implemented all of your valuable feedback in our latest revision, and we would greatly appreciate it if you could let us know your thoughts on the improvements we made.
>
> Below is a brief overview of how we incorporated your specific feedback.
>
> ---
> ---
>
> - **Quantitative representation shattering measures**
>   - Introduced a metric: $R(D_*) = ||D_* - D_{\varnothing}|| / ||D_{\varnothing}||$.
>   - Consistently report this measure in new results (Figures 5–7, Table 2).
>   - Detailed in Section 3.4.
>
> - **Different models and editing methods**
>   - Added experiments with a larger, non-GPT-like architecture model (Mamba-S4 2.8B, in Appendix F.5.3).
>   - Added experiments with **MEMIT, AlphaEdit, and PMET** (Appendix F.5.1 and F.5.2) and Mamba-S4 (Appendix F.5.3).
>
> - **Improved writing, figures, and captions**
>   - Restructured / re-wrote Section 3 (Formalizing Knowledge Editing)
>   - Fixed typos, simplified confusing figures, and added clear notation definitions
>
> ---
> ---
>
> ### Summary
>
> - In our latest revision, we:
>   - Defined a quantitative metric of representation shattering and updated our results to consistently rely on the metric.
>   - Verified the reproducibility of representation shattering across additional KE methods and models and showed that our findings hold across different approaches and model architectures.
>   - Thoroughly revised our manuscript to clarify confusing points and improve readability.
>
> We hope these improvements address your concerns and merit a higher score to support acceptance!

---

### Official Review · Reviewer_DzjZ · 2024-11-04

**Soundness:** 2
**Presentation:** 3
**Contribution:** 3
**Rating:** 5
**Confidence:** 4

**Summary:**

The paper constructs a synthetic dataset with cyclic knowledge graphs and investigates the influences of knowledge editing on the synthetic datasets. The results show that KE inadvertently affects representations of entities beyond the targeted one, distorting relevant structures that allow a model to infer unseen knowledge about an entity.

**Strengths:**

1, The synthetic dataset is well-designed in a way that all the relations are related in a clear semantic meaning. And Figure 3 clealy demonstrates the structure of the synthetic dataset.

2, The visualization is really good. It clearly shows the representation shattering after the knowledge editing (Figure 4, 5, 6 and 8).

**Weaknesses:**

1, The expected answers (or the ground truth answers) after knowledge editing is not clearly stated in the paper. For example in Figure 3 Ring I, after the editing of $1.I\\_C2 = 3$, is $1.I\\_C2 = 2$ still true (so that entity 1 now have 2 entities for relation I_C2) or does this edge simply disappear? Which one of the following is considered the ground truth for evaluation after the editing: $1.I\\_C3 = 4$ or $1.I\\_C4 = 4$? In the proposed synthetic dataset, it could still have a consistent logic for LLMs even after knowledge editing. The paper should make the true facts after the knowledge editing more clear.

2, The prompt templates and codes are not released. How the query and the structural information of KGs are provided in the prompt will have a big influence on the final experiment results. Can the authors provide them during the rebuttal? And can authors show some examples of the LLM outputs in natural languages after knowledge editing to help readers better understand the mistakes LLMs make?

3, More knowledge editing baselines should be included. In this paper ROME is the mainly considered knowledge editing methods. There are other 'locate and edit' methods such as PMET[1]. And there are other types of knowledge editing with extra parameters such as GRACE[2]. Including more knowledge editing methods will make the 'representation shattering' hypothesis more convincing.

4, Some typos need to be fixed. For example, $(x_i, a, x_j)$ should be  $(x_i, r, x_j)$ in line 191. Why is “1 IC_4 3 IIIA_2 7 IIIA_3 2 IIC_2 6” in line 235-236 a plausible sequence in figure 3? $(x_i lr, x_j)$ is not clear in line 291.

[1] Li, Xiaopeng, et al. "Pmet: Precise model editing in a transformer." Proceedings of the AAAI Conference on Artificial Intelligence. Vol. 38. No. 17. 2024.

[2] Hartvigsen, Tom, et al. "Aging with grace: Lifelong model editing with discrete key-value adaptors." Advances in Neural Information Processing Systems 36 (2024).

**Questions:**

Please refer to the questions mentioned in the Weaknesses part.

---

> ### Author Response · Authors · 2024-11-28
> **Rebuttals (1/2)**
>
> We thank the reviewer for taking the time to give us feedback. We are glad that the reviewer found our synthetic setup to be well-motivated, convincing, and our qualitative results to be helpful in highlighting the phenomenon of representation shattering. We respond to specific comments below.
>
> ------
> ------
>
> ### Main comments
>
> > **The expected answers (or the ground truth answers) after knowledge editing is not clearly stated in the paper. For example in Figure 3 Ring I, after the editing of 1.I_C2 =3, is 1.I_C2=2 still true …  or does this edge simply disappear?**
>
> We refer the reviewer to Section 3, where we detail our data-generating process and clarify how the ground truth for edits are created (for both corrective and counterfactual edits). To summarize, since the knowledge graph is defined by us procedurally, the ground truth corresponding to edits are also procedurally ascertained. That is, given any arbitrary node, following the data-generating process’s rules will help one procedurally arrive at the correct answer for an edit.
>
> To specifically address the example raised by the reviewer, we note the edge (i.e., 1.I_C2=3) simply disappears in this case, and the edge (relation) to 2 no longer exists after the knowledge edit.
>
> ------
>
> > **Which one of the following is considered the ground truth for evaluation after the editing: 1.I_C3=4 or 1.I_C4=4 ? In the proposed synthetic dataset, it could still have a consistent logic for LLMs even after knowledge editing. The paper should make the true facts after the knowledge editing more clear.**
>
> This is a great question! To appreciate why this is tricky to define, consider the following example: let us make a counterfactual edit that changes Lionel Messi into a professional basketball player. Did Lionel Messi still play football (in the team Barcelona), or is that no longer true? The challenge of knowledge editing is compounded by the ambiguity in defining what the underlying ground truth should look like and there is no single correct definition.
>
> In our work, the ground truth after a counterfactual edit will be I_C3=3, i.e., the other facts are not changed. While this is not always the right choice of ground truth for all naturalistic scenarios, we adopt this choice for our synthetic KGs. We have now emphasized this in the manuscript.
>
> ------
>
> > **The prompt templates and codes are not released**
>
> We appreciate the reviewer’s comment and emphasize that we take reproducibility seriously. In the manuscript, we describe the prompt templates used for experiments involving real LLMs (i.e., Lines 470-472). As for code, we promise we have a fairly well-written codebase planned for release after the paper is accepted.
>
> ------
>
> > **can authors show some examples of the LLM outputs in natural languages after knowledge editing**
>
> We appreciate the thoughtfulness of this question. Previous works provide representative examples of KE algorithms’ failures with realistic edits in natural language. In particular, we recommend the reviewer explore “Appendix G: Generation Examples” of the ROME manuscript [1]. However, for completeness we describe below an example of a failure from our counterfactual calendar month editing task with GPT-2.
>
> **Setup.** In this example, the unedited model initially correctly outputs that one month from February is March. Then, even after applying the counterfactual edit “One months from January is March” (a distance of 1), the model can still correctly reason that one month from February is March. However, after applying the counterfactual edit “One months from January is June” (a distance of 4), the model loses the ability to reason that March follows February (instead outputting “May”).
>
> ```
> ## Unedited Model
> [Pre-ROME]:   One months from February is March and one month from...
>
> ## Counterfactual Edit Distance = 1
> Executing ROME algorithm for the update: [Let's do some calendar math. One months from January is] -> [ March]
> [Post-ROME]:  Let's do some calendar math. One months from February is March. So we can...
>
> ## Counterfactual Edit Distance = 4
> Executing ROME algorithm for the update: [Let's do some calendar math. One months from January is] -> [ June]
> [Post-ROME]:  Let's do some calendar math. One months from February is May, and one month...
> ```
>
> We hope these pointers and examples above are helpful! We would also be happy to include these examples in the appendix in the final version of the paper.
>
> [1] Kevin Meng, David Bau, Alex Andonian, and Yonatan Belinkov. Locating and editing factual
> associations in GPT. Advances in Neural Information Processing Systems, 35, 2022a.
>
> ------
> ------
> **[Continued below...]**

---

> ### Author Response · Authors · 2024-11-28
> **Rebuttals (2/2)**
>
> > **More knowledge editing baselines should be included. In this paper ROME is the mainly considered knowledge editing methods. There are other 'locate and edit' methods such as PMET[1].**
>
> Thank you for this suggestion. In addition to ROME, we have added new experiments with **three other knowledge editing methods**: namely MEMIT, AlphaEdit, and PMET. We feature these methods in Appendix F.5.1 and F.5.2. We find that the newer editing methods generally degrade the model’s accuracy to a lesser extent than ROME. However, corrective edits still degrade accuracy and counterfactual edits cause greater drops in accuracy for greater distances, i.e., **representation shattering is seen across all these knowledge editing methods.**
>
> ------
>
> > **Why is “1 IC_4 3 IIIA_2 7 IIIA_3 2 IIC_2 6” in line 235-236 a plausible sequence in figure 3?**
>
> Thank you for pointing this out! There was a typo here, and the correct plausible sequence is
> 1 IC_4 4  III_A2 8 II_ A3 3 II_C2 7. The above is a valid sequence because each fact is valid according to Figure 3. In particular  (1 IC_4 4) is valid since the clockwise  4-hop neighbor of entity 1 in order I is entity 4. Similarly, the  (4  III_A2 8) is valid since the anti-clockwise 2-hop neighbor of entity 4 in cyclic order III is entity 8. We can similarly reason about all the other facts in the sequence.
>
> ------
>
> > **Some typos need to be fixed…**
>
> Thank you for pointing them out. We have fixed the typos line 191 and line 291.
>
> ---
> ---
>
> ### Summary
>
> We thank the reviewer for their feedback that has helped us substantially improve the clarity of our paper. Specifically, we have reworked Section 3 improve the clarity of our data-generation process, and also edited several other parts of the paper to emphasize the experimental protocol (e.g., adding prompt templates); these changes are marked in blue in the updated manuscript. We hope our updates help address the reviewer's concerns, and, if so, that they would consider raising their score to support our work's acceptance!

---

> ### Author Response · Authors · 2024-12-01
> **Discussion Period Reminder**
>
> Dear Reviewer,
>
> We thank you again for your detailed feedback on our work. Given the discussion period ends soon, we wanted to check in if our provided responses address your concerns, and see if there are any further questions that we can help address.
>
> Thank you!

---

> ### Author Response · Authors · 2024-12-02
> **Follow-up Reminder**
>
> Dear reviewer,
>
> This is a gentle reminder that the discussion period is ending soon. We have implemented all of your valuable feedback in our latest revision, and we would greatly appreciate it if you could let us know your thoughts on the improvements we made.
>
> Below is a brief overview of how we incorporated your specific feedback.
>
> ---
> ---
>
> - **Additional KE methods**
>   - Added experiments with **MEMIT, AlphaEdit, and PMET** (Appendix F.5.1 and F.5.2) and Mamba-S4 (Appendix F.5.3).
>   - Findings: Newer methods degrade model accuracy less than ROME, but representation shattering persists across all methods.
>
> - **Improved writing and documentation of details**
>   - Described prompt templates and experimental details
>   - Responded to questions regarding counterfactual editing
>   - Provided an example of post-edit model failures
>   - Clarified commitment to releasing our codebase publicly if the paper is accepted
>   - Fixed typos and re-wrote text to improve readability
>
> ---
> ---
>
> ### Summary
>
> - In our latest revision, we:
>   - Verified the reproducibility of representation shattering across additional KE methods and models and showed that our findings hold across different approaches and model architectures.
>   - Clarified our commitment to publishing relevant code if accepted.
>   - Thoroughly revised our manuscript to clarify confusing points and improve readability.
>
> We hope these improvements address your concerns and merit a higher score to support acceptance!

---

### Official Review · Reviewer_Ek2y · 2024-11-04

**Soundness:** 2
**Presentation:** 2
**Contribution:** 2
**Rating:** 5
**Confidence:** 2

**Summary:**

This paper provides a way to investigate the representation scattering hypothesis during knowledge editing. It proposed a test for using synthetic entities to probe if the representation scattering happens.

**Strengths:**

* A new hypothesis, representation scattering,  is proposed
* Experiments are done to test the hypothesis proposed,

**Weaknesses:**

* Only the GPT model is studied, which makes the scope limited in the model architecture.
* The result seems hard to reproduce.

**Questions:**

* Is the assumption still valid on different model architectures, including encoder-decoder models like T5, and Mamba?
* Is the assumption still valid on larger LLM models like Lamma, Gemma, Phi 3.5 etc?

---

> ### Author Response · Authors · 2024-11-28
> **Rebuttals (1/1)**
>
> We thank the reviewer for taking the time to review our work and are glad that the reviewer finds our representation shattering hypothesis to be new. We have addressed all the reviewer’s concerns below and hope to engage in a discussion.
>
> ------
> ------
> ### Main comments
>
> > **Is the assumption still valid…**
>
> We emphasize that we make no assumptions in our work. Our proposed hypothesis, representation shattering, is a concept that we put forth based on extensive empirical evidence in synthetic and naturalistic settings. In fact, during the course of revisions we have now added several new editing protocols (4 in total), new models (Mamba variants), and new graph structures: **our results generalize across all the novel settings as well!**
>
> ------
>
> > **on different model architectures, including encoder-decoder models like T5, and Mamba?**
>
> It would certainly be interesting to test our hypothesis for other model architectures! To this end, we have now added experiments analyzing the effects of editing Mamba-S4 via ROME (or, specifically, the variant of that algorithm that was specifically designed for editing Mamba-S4 by Sharma et al. [1]). Results are provided in Appendix F.5.3, and **we find (both qualitatively and quantitatively) the same effects as our experiments with GPT-2 models**: edits at a larger distance yield more representation shattering!
>
> [1] https://arxiv.org/abs/2404.03646
>
> ------
>
> > **Is the assumption still valid on larger LLM models like Lamma, Gemma, Phi 3.5 etc?**
>
> Thank you for the suggestion. In our aforementioned experiment with Mamba-S4, we used the Mamba-2.8B variant to scale up beyond GPT-2. Our results show that representations of concepts follow similar geometries regardless of model size. Furthermore, we find that the extent of representation shattering increases with the edit distance as expected.
>
> ------
>
> > **The result seems hard to reproduce.**
>
> We appreciate the reviewer’s comment and emphasize that we take reproducibility seriously. We promise we have a fairly well-written codebase planned for release after the paper is accepted.
>
> ---
> ---
>
> ### Summary
>
> We thank the reviewer for their feedback that has helped us improve the robustness of our claims. Specifically, we have added experiments with a **new model architecture** to address reviewer's comment. We note *we have also added experiments with 3 more editing protocols and a different graph structure*, which may interest the reviewer. We hope these updates help address the reviewer's concerns, and, if so, that they would consider raising their score to support our work's acceptance!

---

> ### Author Response · Authors · 2024-12-01
> **Discussion Period Reminder**
>
> Dear Reviewer,
>
> We thank you again for your detailed feedback on our work. Given the discussion period ends soon, we wanted to check in if our provided responses address your concerns, and see if there are any further questions that we can help address.
>
> Thank you!

---

> ### Author Response · Authors · 2024-12-02
> **Follow-up Reminder**
>
> Dear reviewer,
>
> This is a gentle reminder that the discussion period is ending soon. We have implemented all of your valuable feedback in our latest revision, and we would greatly appreciate it if you could let us know your thoughts on the improvements we made.
>
> Below is a brief overview of how we incorporated your specific feedback.
>
> ---
> ---
>
>
> - **Different models and editing methods**
>   - Added experiments with a larger, non-GPT-like architecture model (Mamba-S4 2.8B, in Appendix F.5.3).
>   - Added experiments with **MEMIT, AlphaEdit, and PMET** (Appendix F.5.1 and F.5.2) and Mamba-S4 (Appendix F.5.3).
>
> - **Clarified commitment to reproducibility**
>   - We will release our codebase publicly if the paper is accepted.
>
> ---
> ---
>
> ### Summary
>
> - In our latest revision, we:
>   - Verified the reproducibility of representation shattering across additional KE methods and models and showed that our findings hold across different approaches and model architectures.
>   - Clarified our commitment to publishing relevant code if accepted.
>
> We hope these improvements address your concerns and merit a higher score to support acceptance!

---

### Official Review · Reviewer_DH8A · 2024-11-05

**Soundness:** 2
**Presentation:** 2
**Contribution:** 4
**Rating:** 5
**Confidence:** 3

**Summary:**

The key idea of the paper is that knowledge editing (KE) alters the geometry of transformer models' representations, making them inconsistent or "shattered." The authors introduce the concept of "representation shattering" to explain why KE results in degradation of the model's performance. This perspective of geometric shattering is both insightful and novel, providing a deeper understanding of how targeted edits can inadvertently disrupt broader model behavior. The study is conducted using synthetic knowledge graphs (KGs), which allows the authors to identify specific geometric changes in transformer representations.

**Strengths:**

1. **Novelty**: This paper is the first to delve deeply into transformers' representations, providing a mechanistic understanding of why the detrimental impacts of KE occur. In contrast, previous work mainly treats transformers and LLMs as black boxes, focusing on observing the degradation of models' overall performance in targeted tasks.
2. **Unique Approach and Insightful Perspective**: The authors tackle the problem from the perspective of studying representation geometry, which offers a unique and insightful angle into how KE affects model consistency.
3. **Preliminary Results on LLM**: The authors extend their findings to a pretrained large model, GPT-2-XL, showing that the representation shattering phenomenon also applies to LLMs, not just to transformers trained on synthetic data.

**Weaknesses:**

Although the perspective and hypothesis of this paper are novel and insightful, I have concerns regarding the limitations of the experimental setup, specifically the reliance on cyclic structure in the synthetic KGs and the focus on only one KE method.

1. **Why Cyclic Graphs?** The key assumption of this study is that transformers can learn a cyclic geometric representation when the training KGs have cyclic relation subgraphs. This assumption on its own makes sense. However, to my knowledge, real-world KGs often do not have relation subgraphs that conform to cyclic orders. For example, take the advisor relation that is used as the running example in the paper. It is unlikely that "Alice is the advisor of Bob", "Bob is the advisor of Charlie", and "Charlie is the advisor of Alice." In reality, most KG relations seem to form trees instead of cycles. This makes the study somewhat disconnected from real-world scenarios as cyclic relations are uncommon. Why not investigate the geometry of transformer representations in real-world KGs (or a semi-synthetic KG that more closely mimics the structure of real-world KGs) when the relation subgraphs are trees, and examine how KEs affect such geometry?
2. **Knowledge Graph Construction**: The construction of the knowledge graphs, particularly the use of cycles, is not very intuitive and is challenging to understand. This is compounded by conflicting information in Section 3.2 (e.g., Line 202 states that the KG is defined to have 3 relations, but Line 205 later refers to 24 relations) and an explanation that omits critical details. Specifically, it is unclear what the exact cyclic orders are for the edit, retain, and test relations, and whether these relations are organized in such a way that each of the three groups has distinct orders. Additionally, Figure 3, intended to provide more intuition and details, is difficult to interpret due to unclear labeling, such as the meaning of different arrow colors and which edges were edited. Clarifying why the authors chose this specific construction, particularly the necessity of the use of different cyclic orders, and providing more transparent explanations of the data construction would greatly improve the reader's ability to understand the motivation behind the experimental setup.
3. **Single KE Method (ROME)**: The paper only studies one knowledge editing method, ROME. The findings would be significantly strengthened if multiple KE methods were analyzed to determine whether the representation shattering phenomenon is a general property of transformer models occurring on a more fundamental representation level, independent of the KE method used. For instance, the paper shows that shattering occurs even for corrective edits, which is surprising because a perfect corrective edit should ideally refine the model's representation geometry. It is thus crucial to investigate whether this surprising phenomenon is specific to ROME or is a broader theme across different KE methods.
4. **Lack of Mitigation Strategies**: While the paper does an excellent job of identifying the issue of representation shattering, it lacks discussion on potential ways to mitigate this problem. Including some hypotheses or suggestions for future work would provide direction for researchers interested in addressing this issue.

In addition, some other comments on readability and presentation:

5. **Figures**: The Figure 1 is somewhat confusing, particularly in panel (b). It is unclear how the ring structure is represented in the visualization, and what the labels "182" and "608" refer to. Additionally, Figure 1 is not mentioned in the introduction, which makes it difficult to understand its relevance early on. It would help if the illustration was used to convey the high-level intuition of the findings. Similarly for Figure 3, mentioned in W2.
6. **Undefined Terms**: The abbreviation "DGP" (Figure 1 caption, line 67) is undefined. I assume it stands for "Data Generating Process"—clarifying this would improve readability.
7. **Entity and Relation Naming**: The naming conventions for entities and relations are unclear. For instance, on line 235, it is not immediately evident that "1," "3," etc., are entities, and that "I_C4," "III_A2," etc., are relations. In addition, there are clues in Figure 3 that the number of "I"'s stand for the different cyclic order, but this is not mentioned and explained in the main text. An explanation of the naming convention and its significance (e.g., whether the number of "I"s corresponds to hops in the KG) would be helpful.
8. **Typo**: On line 291, the phrase "it changes fact $(x_i l r, x_j)$" should probably be "it changes fact $(x_i, r, x_j)$"
9. **Unclear Sentence**: On line 366, the caption of Figure 5 reads: "We plot the mean drop in accuracy against the in the Frobenius norm of the difference in..." This sentence does not make sense as written and needs to be revised.

**Questions:**

1. **Possibility of Studying Geometry on Real-World KGs**: As previously raised in W1, would it be feasible to apply the same methodology to real-world KGs to examine how the representation geometry shatters under KEs? Would the lack of cyclic orders make this significantly more challenging? Additionally, could the Isomap visualization technique still be used effectively in this scenario to visualize a tree-like structure rather than rings?
2. **Training Setup Clarification**: Are all edit, retain, and test relations used during the pre-training of the transformer, and the test relations are only held out specifically for the KE process but they are all observed by the transformer during pre-training? If so, a clarification near line 213 would be helpful.
3. **Logical Inference Task**: For the logical inference task, could you elaborate on how the "hold out" process works? For example, if the inverse relation (say "advisee") is held out and unobserved during training, how can the model know that the held-out relation token "advisee" is the inverse of the observed relation "advisor"?

---

> ### Author Response · Authors · 2024-11-28
> **Rebuttals (1/2)**
>
> We thank the reviewer for their feedback and for taking the time to review our work! We are glad that the reviewer felt our mechanistic study on knowledge editing offers a unique perspective from the lens of representation geometry. We have added additional experiments which hopefully convince the reviewer that representation shattering extends beyond cyclic structures and to many other knowledge editing methods.
>
> ------
> ------
> ### Main comments
>
> > **Why not investigate the geometry of transformer representations in real-world KGs when the relation subgraphs are trees, and examine how KEs affect such geometry?**
>
> Mathematically describing what knowledge graphs look like in the real world is difficult. However, inspired by your suggestion to explore real-world knowledge graphs consisting of trees, we have added preliminary experiments with a **knowledge graph of countries and cities represented as a small tree** using a real language model (GPT-2). **The results align with our main findings with our synthetic KGs**: KE methods distort language models' representations in order to insert new facts or alter old ones (i.e., representation shattering), and the extent of representation shattering increases with the distance between the old fact and the desired new fact on the manifold. Furthermore, we found that counterfactual edits essentially collapse the representations of the countries and cities. For a detailed exploration, please kindly refer to Appendix F.6.
>
> ------
>
> > **The paper only studies one knowledge editing method, ROME. The findings would be significantly strengthened if multiple KE methods were analyzed**
>
> Thank you for this suggestion. In addition to ROME, we have added new experiments with **three other knowledge editing methods**: namely MEMIT, AlphaEdit, and PMET. We feature these methods in Appendix F.5.1 and F.5.2. We find that the newer editing methods generally degrade the model’s accuracy to a lesser extent than ROME. However, corrective edits still degrade accuracy and counterfactual edits cause greater drops in accuracy for greater distances, i.e., **representation shattering is seen across all these knowledge editing methods.**
>
> ------
>
> > **The construction of the knowledge graphs, particularly the use of cycles, is not very intuitive and is challenging to understand**
>
> We have significantly improved the writing in Section 3, with the salient changes highlighted in blue. In particular, we have added a more detailed description of the setup and the reasoning behind using 3 cyclic orders. We have addressed specific questions below.
>
> ------
>
> > **what the exact cyclic orders are for the edit, retain, and test relations...**
>
> The 3 orders are defined over the 2048 entities and are sampled at random. All three orders are different from each other. This detail is now included in Section 3.2.
>
> ------
>
> > **why the authors chose this specific construction, particularly the necessity of the use of different cyclic orders...**
>
> Thank you for raising this point! We have significantly expanded the text to address this question (see paragraph starting at Line 199). To summarize, we edit a relation that is associated with the first order. We need a second cyclic ordering of the entities to define a set of facts that remain unchanged after an edit. These facts are used by knowledge editing algorithms like ROME. We additionally define a third cyclic order which corresponds to facts that were neither edited, nor used by the knowledge editing algorithm. These serve as a held out set of facts for evaluation.
>
> ------
>
> > **it lacks discussion on potential ways to mitigate this problem. Including some hypotheses or suggestions for future work.**
>
> This is a great question! However, we emphasize that the goal of our paper was to establish a mechanistic hypothesis for why knowledge editing protocols impact model capabilities. As we show, preserving representational structures underlying a model’s knowledge is likely crucial to avoid negative consequences of knowledge editing: distortion of such structures impacts a model’s broader capabilities. While a concrete answer to the question of how to design a knowledge editing protocol that can accommodate these structures is out of scope of our work, we believe that such a protocol should at least perform well on our synthetic knowledge graph setup. Failing even this simple, albeit systematically defined setting, likely implies the protocol should not be readily trusted or applied at scale. Following your recommendation, we have added a short discussion of these recommendations to Section 5 (Conclusion).
>
>
> ------
> **[Continued below...]**

---

> ### Author Response · Authors · 2024-11-28
> **Rebuttals (2/2)**
>
> > **Additionally, could the Isomap visualization technique still be used effectively in this scenario to visualize a tree-like structure rather than rings?**
>
> Yes! Isomap constructs the geometry of the manifold using a local neighborhood graph and is not designed specifically for cycles: it can also be used with other structures, including ones that are tree-like. To demonstrate this, we have now added new experiments in Appendix F.6 where we show isomaps continue to work well beyond ring structures.
>
> ----
>
> > **For the logical inference task, could you elaborate on how the "hold out" process works?**
>
> We refer the reviewer to Appendix B, where we describe in detail the hold-out process. To summarize, for the logical inference task, any facts about an entity’s clockwise neighbors can be inferred if it is known that the said entity is an anti-clockwise neighbor of another entity. For example, even if “Bob is the advisee of Alice” is held out during training, the fact “Alice is the advisor of Bob” can be inferred, so long as both relation directions are observed for other pair of individuals in pre-training (i.e., “Carol is the advisor of David” and “David is the advisee of Carol”). As we elaborate in Appendix B, we only show both directions of neighboring relations with a probability of 1/3rd. The model must use this 1/3rd fraction of the relations to infer the remaining 2/3rds of the facts.
>
> ---
> ---
>
> ### Suggested improvements to writing
>
> > **The Figure 1 is somewhat confusing, particularly in panel (b)**
>
> Thank you for pointing this out. To further clarify the figure, we have added additional annotations and expanded the caption. In brief, we note the bar graph in panel (b) represents the probability of generating different entities. The bar graph below is a probability distribution over the tokens and highlights that the edit changes the answer for “the neighbor to the right of entity #123”. However, this edit also degrades the model’s accuracy on direct recall, logical inference, and compositional inference.
>
> ------
>
> > **in Section 3.2 (e.g., Line 202 states that the KG is defined to have 3 relations, but Line 205 later refers to 24 relations)**
>
> Thank you for pointing out this typo. We have fixed this in the updated manuscript. To clarify, the knowledge graph has 3 cyclic orders which are used to define 24 relations.
>
> ------
>
> > **Figure 3, intended to provide more intuition and details, is difficult to interpret due to unclear labeling, such as the meaning of different arrow colors and which edges were edited**
>
> We apologize if the labelling was unclear. We have added more details to the caption and improved the quality of the Figure to make it easier to interpret. Figure 3 (top row) depicts an edit that changes the red arrow to the green arrow. The distance of the edit is 1, which is defined with respect to cyclic order I.
>
> ------
>
> > **The abbreviation "DGP" (Figure 1 caption, line 67) is undefined.**
>
> We apologize for this! DGP stands for data generating process; we have fixed it in the latest version of the manuscript.
>
> ------
>
> > **An explanation of the naming convention and its significance (e.g., whether the number of "I"s corresponds to hops in the KG) would be helpful.**
>
> We have added another paragraph to the main text explaining the notation and more prominently featured the text in the caption of Figure 3. In particular, I, II, and III are used to represent the 3 different cyclic orders. The entities are named by the ID number (1-2048). The relations are named using a combination of the cyclic order (I, II, III), the neighbor's distance (1-4), and the neighbor's direction (Clockwise / Anti-clockwise). For instance, the relation “I_C2” is the 2-hop neighbor in the clockwise direction.
>
> ------
>
> > **Typo On line 291, the phrase "it changes fact"**
>
> Thank you for pointing this out. We have fixed it in the newer version of the manuscript.
>
> ------
>
> > **Unclear Sentence: On line 366, the caption of Figure 5 reads: "We plot the mean drop in ac...**
>
> We have removed the text “in the”, from the sentence. Thank you for pointing out this typo.
>
> ------
>
> > **Are all edit, retain, and test relations used during the pre-training of the transformer, and the test relations are only held out specifically for the KE process**
>
> Yes, all 3 relations are seen during pre-training to ensure the model internalizes these relations (else, editing it would be impossible). To make this clearer, we have added more discussion in Lines 216-217 as well as Lines 237-242.
>
> ---
> ---
>
> ### Summary
>
> We thank the reviewer for their feedback that has helped us improve the clarity of our paper and the robustness of our claims. Specifically, we have added experiments with **3 new editing methods and a different graph structure**, along with *several writing edits* to improve clarity. We hope these updates help address the reviewer's concerns, and, if so, that they would consider raising their score to support our work's acceptance!

---

> ### Author Response · Authors · 2024-12-01
> **Discussion Period Reminder**
>
> Dear Reviewer,
>
> We thank you again for your detailed feedback on our work. Given the discussion period ends soon, we wanted to check in if our provided responses address your concerns, and see if there are any further questions that we can help address.
>
> Thank you!

---

> ### Author Response · Authors · 2024-12-02
> **Follow-up Reminder**
>
> Dear reviewer,
>
> This is a gentle reminder that the discussion period is ending soon. We have implemented all of your valuable feedback in our latest revision, and we would greatly appreciate it if you could let us know your thoughts on the improvements we made.
>
> Below is a brief overview of how we incorporated your specific feedback.
>
> ---
> ---
>
>
> - **Realistic graph structures (trees)**
>   - Conducted experiments with a small tree-structured KG (countries and cities).
>   - Results support our findings: KE methods distort representations (representation shattering), with greater shattering at larger fact distances. See Appendix F.6.
>
> - **Additional KE methods**
>   - Added experiments with **MEMIT, AlphaEdit, and PMET** (Appendix F.5.1 and F.5.2) and Mamba-S4 (Appendix F.5.3).
>   - Findings: Newer methods degrade model accuracy less than ROME, but representation shattering persists across all methods.
>
> - **Improved writing, figures, and captions**
>   - Clarified the graph construction and data generation processes
>   - Restructured / re-wrote Section 3 (Formalizing Knowledge Editing)
>   - Fixed typos, simplified confusing figures, and added term definitions
>   - Added discussion of recommendations for KE methods (Section 5)
>
> ---
> ---
>
> ### Summary
>
> - In our latest revision, we:
>   - Further verified the hypothesis by extending the synthetic setup to incorporate small, realistic, tree-shaped KGs.
>   - Verified the reproducibility of representation shattering across additional KE methods and showed that our findings hold across different approaches.
>   - Thoroughly revised our manuscript to clarify confusing points and improve readability.
>
> We hope these improvements address your concerns and merit a higher score to support acceptance!

---

### Official Review · Reviewer_jMAi · 2024-11-08

**Soundness:** 2
**Presentation:** 3
**Contribution:** 2
**Rating:** 3
**Confidence:** 3

**Summary:**

The paper introduces the Representation Shattering Hypothesis, which posits that Knowledge Editing (KE) methods distort a model's latent representations, potentially to the point of losing the learned global graph structure—a phenomenon they call "representation shattering." The authors hypothesize that this mechanism underlies KE's negative impact on models' factual and reasoning abilities.

To test this hypothesis, they first create a toy knowledge graph to pre-train a language model on. Then, using a popular KE method called ROME, they edit facts within this graph, demonstrating that the extent of representation distortion correlates with how significantly an edit disrupts the graph's structure.

The authors further validate their hypothesis with a real language model, GPT2, by using ROME to corrupt the order of the months of the year. They show that this corruption distorts the ring structure of the month representations, with greater distortions occurring for larger edit distances.

**Strengths:**

- **Insight into Adverse Effects of KE:** With knowledge editing (KE) methods gaining popularity, this paper provides valuable insights into why KE can lead to adverse effects, offering a deeper understanding of its impact on model representations.

- **Clarity and Accessibility:** The paper is well-written and accessible, making complex concepts easier to understand for a broad audience.

- **Novel Toy Knowledge Graphs:** The authors introduce innovative toy knowledge graph (KG) tasks to explore KE and the phenomenon of representation shattering, providing a structured and interpretable framework for analyzing KE effects.

**Weaknesses:**

### Weaknesses

- **Limited Scope of Experimentation:**
  - The authors primarily experiment on knowledge graphs (KGs) with a ring-like structure, which is a somewhat artificial setup. Even the experiment with a real KG is restricted to a small subset with a similar ring configuration, limiting the generalizability of the findings.
  - Only one knowledge editing (KE) technique, ROME, is tested, leaving it unclear how the findings might extend to other KE methods.

- **Reliance on Visual Inspection of Representation Shattering:**
  - The approach seems heavily dependent on visually inspecting the breakdown of representations, which becomes infeasible for more complex, unvisualizable data. Although the authors propose using the Frobenius norm as a quantitative measure of shattering, it is only applied in one experiment.

### Suggestions for Improvement

- Extend the synthetic setup to incorporate small, realistic KGs to further validate the hypothesis.
- Experiment with additional KE methods to assess whether the representation shattering phenomenon holds across different approaches.
- Rely more consistently on quantitative measures (like maybe the Frobenius norm) to gauge representation shattering, especially in cases where visual inspection may not be feasible.

**Questions:**

see weaknesses

---

> ### Author Response · Authors · 2024-11-28
> **Rebuttals (1/1)**
>
> We thank the reviewer for their valuable feedback and for taking the time to review our work! We are glad the reviewer finds that our work has valuable insights for knowledge editing, the tasks to be innovative and the paper to be well-written and accessible. **We have run new experiments using other knowledge editing methods and tree-structured graphs**, which we hope sufficiently addresses the reviewer’s concerns.
>
> ------
> ------
> ### Main comments
>
> > **Extend the synthetic setup to incorporate small, realistic KGs to further validate the hypothesis.**
>
> Inspired by your suggestion to explore small, realistic knowledge graphs, we conducted preliminary experiments with a knowledge graph of countries and cities represented as a small tree using a real language model (GPT-2). *The results align with our main findings arrived at using our synthetic KGs:* KE methods distort language models' representations in order to insert new facts or alter old ones (i.e., representation shattering), and the extent of representation shattering increases with the distance between the old fact and the desired new fact on the manifold. For more details, see Appendix F.6.
>
> ------
>
> > **Only one knowledge editing (KE) technique, ROME, is tested…**
>
> Thank you for this suggestion. In addition to ROME, we have added new experiments with **three other knowledge editing methods**: namely MEMIT, AlphaEdit, and PMET. We feature these methods in Appendix F.5.1 and F.5.2. We find that the newer editing methods generally degrade the model’s accuracy to a lesser extent than ROME. However, corrective edits still degrade accuracy and counterfactual edits cause greater drops in accuracy for greater distances, i.e., **representation shattering is seen across all these knowledge editing methods.**
>
> ------
>
> > **Rely more consistently on quantitative measures**
>
> Thank you for this suggestion. We have added a new section (3.4) on representation shattering that presents a quantitative measure of representation shattering. In particular, we use $R(D_*) = ||D_* - D_{\varnothing}|| / ||D_{\varnothing}||$, where the norm is the Frobenius norm, $D_{\varnothing}$ is the pairwise distances between the representations of entities of the unedited model and $D_*$ is the pairwise distances between representations of the edited model. **We now consistently report $R(D_*)$ in our experiments (Figures 5, 6, 7, Table 2).**
>
> ---
> ---
>
> ### Summary
>
> We thank the reviewer for their feedback and suggestions which have helped improve the robustness of our claims. In particular, we have now added experiments with (i) **3 new editing methods**, (ii) **a new model architecture (Mamba)**, and (iii) **a different graph structure**, along with (iv) a **thorough quantification of the degree of representation shattering** throughout the paper. We hope these updates to the manuscript help address the reviewer's concerns, and, if so, that they would consider raising their score to support our work's acceptance!

---

> ### Author Response · Authors · 2024-12-01
> **Discussion Period Reminder**
>
> Dear Reviewer,
>
> We thank you again for your detailed feedback on our work. Given the discussion period ends soon, we wanted to check in if our provided responses address your concerns, and see if there are any further questions that we can help address.
>
> Thank you!

---

> ### Author Response · Authors · 2024-12-02
> **Follow-up Reminder**
>
> Dear reviewer,
>
> This is a gentle reminder that the discussion period is ending soon. We have implemented all of your valuable feedback in our latest revision, and we would greatly appreciate it if you could let us know your thoughts on the improvements we made.
>
> Below is a brief overview of how we incorporated your specific feedback.
>
> ---
> ---
>
>
> - **Realistic graph structures (trees)**
>   - Conducted experiments with a small tree-structured KG (countries and cities).
>   - Results support our findings: KE methods distort representations (representation shattering), with greater shattering at larger fact distances. See Appendix F.6.
>
> - **Additional KE methods**
>   - Added experiments with **MEMIT, AlphaEdit, and PMET** (Appendix F.5.1 and F.5.2) and Mamba-S4 (Appendix F.5.3).
>   - Findings: Newer methods degrade model accuracy less than ROME, but representation shattering persists across all methods.
>
> - **Quantitative representation shattering measures**
>   - Introduced a metric: $R(D_*) = ||D_* - D_{\varnothing}|| / ||D_{\varnothing}||$.
>   - Consistently report this measure in new results (Figures 5–7, Table 2).
>   - Detailed in Section 3.4.
>
> ---
> ---
>
> ### Summary
>
> - In our latest revision, we:
>   - Further verified the hypothesis by extending the synthetic setup to incorporate small, realistic, tree-shaped KGs.
>   - Verified the reproducibility of representation shattering across additional KE methods and showed that our findings hold across different approaches.
>   - Defined a quantitative metric of representation shattering and updated our results to consistently rely on the metric.
>
> We hope these improvements address your concerns and merit a higher score to support acceptance!

---

### Author Response · Authors · 2024-11-28
**General Response**

We thank all reviewers for taking the time to provide feedback and engage in the review process. Our manuscript analyzes model representations before and after knowledge editing, which the reviewers found to be valuable in several aspects. The reviewers noted that our writing is "well-written and accessible" (R-jMAi) and our "visualization is really good" (R-DzjZ). They considered our approach with synthetic data to be "innovative" (R-jMAi) and "well-designed" (R-DzjZ), "providing a structured and interpretable framework" (R-jMAi). Additionally, our work was recognized as "the first to delve deeply into transformer's representations" and analyze the "impacts of KE" (R-DH8A).

We have incorporated feedback from the reviews into the manuscript with salient changes highlighted in blue. We have also added 2 new sections to the appendix: Appendix F.5 and F.6. We summarize some of the changes below. Overall, we believe that these changes address many of the common concerns raised by the reviewers.

## New results with different editing protocols, Mamba, and tree-like graphs
The first concern was that representation shattering was only shown with one knowledge editing method. In sections F.5.1, and F.5.2, we have added new experiments with **3 other knowledge editing methods.** We also ran **experiments with a MAMBA-2.8B model** (in Appendix F.5.3), which uses a different architecture and has significantly more parameters than GPT-2XL. The reviewers also raised the concern that our experiments were limited to cyclic graphs. We have conducted **new experiments on tree-like knowledge graphs** that consist of cities and countries as entities (see Appendix F.6). We again find evidence of representation shattering. Furthermore, edits that use entities that are further away in the representation manifold lead to greater degree of representation shattering. **In all our experiments, we found that representation shattering occurs in all these scenarios with the extent of shattering increasing with the distance of the edit.**

## Updated experimental details
To improve the clarity of our work, we have **added more details** to the description of the data generating process and the **construction of the knowledge graphs** (highlighted in blue). We have also improved captions and annotations in some figures as suggested by the reviewers. In order to ensure that the visualizations are faithful, we have **added quantifications of representation shattering to our figures with a well-defined metric.**

## Summary
Overall, we hope that these changes address the primary concerns of the reviewers, and we look forward to engaging in a discussion. We have also responded to individual comments below and urge the reviewers to peruse through the revised version of the manuscript.

---

### Meta-Review · Area_Chair_DDct · 2024-12-19

**Metareview:**

This paper presents a new hypothesis called "Representation Shattering" to explain performance degradation in knowledge editing approaches. The research suggests that knowledge editing can distort a model's latent representations and potentially destroy learned global graph structures. The severity of this distortion appears to correlate with how significantly an edit disrupts the underlying structure, as demonstrated through experiments with both synthetic knowledge graphs and real language models.

The research makes valuable contributions through its novel theoretical framework, which provides a mechanistic explanation for knowledge editing's negative effects rather than treating transformers as black boxes. The methodology is particularly innovative, employing synthetic knowledge graphs for controlled investigation of how edits affect representation geometry. The findings are supported through both qualitative visualization and quantitative metrics, with the results extending from synthetic settings to real models, suggesting this phenomenon may be fundamental to transformer architectures.

However, the research faces several limitations in its current form. The experimental scope is notably restricted, focusing primarily on cyclic structures in knowledge graphs and testing with a single editing method. The methodology heavily relies on visual inspection of representation shattering, with unclear ground truth specifications after editing and insufficient quantitative metrics for measuring distortion. Additionally, the presentation could be improved, particularly in explaining knowledge graph construction and defining key concepts earlier in the paper.

The paper would benefit from broader validation across multiple editing methods and diverse model architectures, along with investigation of non-cyclic knowledge structures. Technical improvements should include more rigorous quantitative metrics for measuring representation shattering, clearer specification of ground truth post-editing, and comprehensive code release for reproducibility. The presentation would be enhanced by introducing key definitions earlier and improving figure clarity and labeling.

While the core concept shows promise and could significantly impact the field, the current submission requires substantial revisions to address these methodological limitations and strengthen the empirical validation of the hypothesis. The recommendation is to reject the paper in its current form, allowing for these necessary improvements to be made before reconsideration.

**Additional Comments On Reviewer Discussion:**

The review process highlighted several key methodological concerns regarding knowledge editing methods and model architecture coverage. Reviewers emphasized the importance of expanding beyond a single knowledge editing approach and testing across different model architectures to ensure broader applicability of the findings.

Questions were raised about the reliance on cyclic knowledge graph structures, with suggestions to explore more realistic tree-like configurations that better reflect real-world relationships. Reviewers also requested clearer documentation of experimental procedures, particularly regarding ground truth answers after editing and the specific prompts used.

In response, the authors conducted extensive additional experiments incorporating multiple knowledge editing methods, which consistently demonstrated representation shattering effects. They expanded their architectural coverage by including experiments with newer models, showing that the phenomenon extends beyond traditional transformer architectures. New experiments using tree-like knowledge graphs with real-world entities demonstrated that representation shattering occurs in non-cyclic structures as well.

The documentation was significantly enhanced with detailed explanations of the data generation process and knowledge graph construction. Visualization improvements included better figure annotations and new quantitative metrics for measuring representation shattering. These changes made the work more accessible and reproducible. However, some limitations persist, including the lack of publicly available code and prompt templates, limited testing on bidirectional models, and room for more extensive real-world knowledge graph experiments.

---

### Decision · Program_Chairs · 2025-01-22

Reject